# Selenoprotein deficiency disorder predisposes to aortic aneurysm formation

Erik Schoenmakers [1,23], Federica Marelli[2,23], Helle F. Jørgensen [3,23], W. Edward Visser [4,23], Carla Moran [1,23], Stefan Groeneweg[4], Carolina Avalos[5], Sean J. Jurgens [6,7], Nichola Figg[3], Alison Finigan[3], Neha Wali[8], Maura Agostini[1], Hannah Wardle-Jones[8], Greta Lyons[1], Rosemary Rusk[9], Deepa Gopalan[10], Philip Twiss[11], Jacob J. Visser[12], Martin Goddard[13], Samer A. M. Nashef[14], Robin Heijmen[15], Paul Clift [16], Sanjay Sinha [3], James P. Pirruccello [6,17], Patrick T. Ellinor [6,18,19], Elisabeth M. Busch-Nentwich[20], Ramiro Ramirez-Solis[8], Michael P. Murphy [21], Luca Persani [2,22], Martin Bennett [3] & Krishna Chatterjee [1] ✉

Aortic aneurysms, which may dissect or rupture acutely and be lethal, can be a part of multisystem disorders that have a heritable basis. We report four patients with deficiency of selenocysteine-containing proteins due to seleno-cysteine Insertion Sequence Binding Protein 2 (*SECISBP2)* mutations who show early-onset, progressive, aneurysmal dilatation of the ascending aorta due to cystic medial necrosis. Zebrafish and male mice with global or vascular smooth muscle cell (VSMC)-targeted disruption of *Secisbp2* respectively show similar aortopathy. Aortas from patients and animal models exhibit raised cellular reactive oxygen species, oxidative DNA damage and VSMC apoptosis. Anti-oxidant exposure or chelation of iron prevents oxidative damage in patient's cells and aortopathy in the zebrafish model. Our observations suggest a key role for oxidative stress and cell death, including via ferroptosis, in mediating aortic degeneration.

Thoracic aortic aneurysms (TAAs) are characterized by weakening of the vessel wall and progressive aortic dilatation and can develop silently and then dissect or rupture with a potentially fatal outcome[1]. TAA occurrence at a young age, in families, or in association with other anomalies (e.g. Marfan, Loeys-Dietz, Ehlers-Danlos type IV syndromes), indicates a strong heritable basis. TAAs frequently exhibit histological changes termed cystic medial necrosis (CMN) or degeneration, characterized by disruption of the media including the elastic laminae, pools of extracellular matrix (ECM) mucopolysaccharide, and loss of vascular smooth muscle cells (VSMCs). The association of pathogenic mutations in genes with both syndromic (e.g. fibrillin 1 *FBN1*, Marfan; transforming growth factor-β (TGFβ) receptors *TGFBR1* and *TGFBR2*, Loeys-Dietz; collagen 3 α-1 *COL3A1*, Ehlers-Danlos IV) and non-syndromic (e.g. smooth muscle myosin, *MYH11*; α-smooth muscle actin, *ACTA2*) TAAs indicates that disordered ECM homoeostasis,

disrupted VSMC function, and perturbed TGFβ signalling are key pathways that contribute to their pathogenesis[2]. However, defects in these and other genes account for only 30% of familial TAA, with the genetic basis of most hereditary TAAs being unknown[3].

Selenium, a trace element essential for human health, is incorporated as the amino acid selenocysteine (Sec) into at least 25 human selenoproteins with diverse biological functions[4]. An evolutionarily conserved mechanism, comprising the interaction of SECISBP2 with specific sequence elements usually located in 3'-untranslated regions of selenoprotein mRNAs, mediates ribosomal incorporation of Sec during synthesis of these proteins[5,6]. Homozygous or compound heterozygous mutations in *SECISBP2* cause deficiency of selenoproteins, resulting in a multisystem disorder (OMIM 607693) with features attributable to deficiencies of tissue-specific selenoproteins (e.g. male infertility, muscular dystrophy), cellular oxidative stress due to loss of

antioxidant selenoenzymes (e.g., glutathione peroxidases, thioredoxin reductases), and abnormal circulating thyroid hormone and selenium levels, reflecting disordered hormone metabolism by deiodinase selenoenzymes and low circulating selenoproteins respectively[7,8].

Although individual (e.g., Selenoprotein P) or subsets (e.g., endoplasmic reticulum resident) of selenoproteins have been linked to adverse cardiovascular outcomes[9] or cardioprotection[10], respectively, they have not been associated with vascular pathology.

Here, we show that oxidative stress-mediated aortic cell death predisposes to thoracic aortic aneurysm formation in humans and animal models with selenoprotein deficiency.

## Results

Patient P1 (male, age 48 years), was born at term, with birth complicated by umbilical cord prolapse and aspiration of amniotic fluid. Birthweight was 2740 g (−1.9 SDS) and length 47 cm (−2.1 SDS), with pronounced muscular hypotonia in the neonatal period. At age 1 year muscular hypotonia, slow psychomotor development and constipation prompted investigation. Although slightly below normal centiles initially, his growth improved such that height and weight gain progressed normally. His bone age was delayed but was in accordance with height age. Thyroid function tests showed raised circulating total T4 concentrations (238 nmol/L, normal range: 60–160), low-normal T3 resin uptake (23.1%; normal range: 22–33%) and normal TSH concentrations (2.9 mU/L, normal range: <1–5). At age 8 years, details of his case, postulating a disorder of unknown aetiology associated with reduced T4 to T3 conversion were reported[11]. At age 19 years sensorineural hearing loss was diagnosed. At age, 23 years, marked widening (8 cm) of the ascending thoracic aorta and severe aortic regurgitation prompted a Bentall procedure (composite valve, root and ascending aorta graft replacement). At age 33 years he developed seizures, requiring levetiracetam therapy. However, at age 35 years he showed aortic arch dilatation distal to the graft replacement (Fig. 1A). At age 42 years a genetic diagnosis of selenoprotein deficiency, with compound heterozygosity for truncation (c.1312A>T; K438X) and mis-splicing/premature stop (c.1894-3C > G; fs631X) SECISBP2 variants (Supplementary Fig. 1A–C), was made. He had raised circulating thyroxine together with low triiodothyronine and selenium levels (Supplementary Fig. 1D). At age 43 years he underwent replacement of the dilated, distal ascending aorta which had progressed inexorably and deficiency of selenoproteins in his aortic vascular smooth muscle cells (VSMCs) was observed (Supplementary Fig. 2A–E). Aortic surveillance (echocardiographic) is undertaken annually.

Patient P2 (male, age 51 years) exhibited delayed milestones (speech, motor) in childhood, requiring speech therapy. Despite myringotomies for otitis media (age 6 years), hearing problems persisted into adulthood when audiometry revealed bilateral sensorineural hearing loss. At age 13 years, he developed marked sun photosensitivity which is ongoing. Following completion of normal growth and pubertal development, complete azoospermia caused primary infertility in adulthood. Progressive weakness, involving multiple muscle groups (neck, paraspinal, limb), causes axial rigidity, reduced mobility and nocturnal hypoventilation, with this phenotype resembling selenoprotein N-deficient myopathy. Symptoms, including fatigue, muscle weakness and digital vasospasm (Raynaud's disease) prompted further investigation leading to identification of biallelic, SECISBP2 variants (c.668delT, c.IVS7-155,T>A; p.F223fs255X,p.fs295X+fs302X) causing selenoprotein deficiency and details of his case upto age 36 years have been reported previously[8]. He developed aortic root dilatation (4 cm) at age 37 years that progressed rapidly (42 years, 4.4 cm; 46 years 4.9 cm; 47 years 5.3 cm) associated with aortic valvular insufficiency (Fig. 1A), requiring aortic root replacement with valve preservation. He had a similar deficiency of selenoproteins in VSMCs (Supplementary Fig. 2A–E). From age 46 years, severe, varicose venous tortuosities in

both lower limbs have required sclerotherapy and endofrequency ablation every two years. Annual echocardiographic surveillance indicates that his aortic root and proximal ascending aorta remain normal in size.

Patient P3 (male, age 25 years) was normal at birth and during early development. Investigation at age 8.9 years for growth retardation showed raised circulating free T4 and low free T3 (FT3) concentrations, leading to the identification of biallelic, pathogenic SECISBP2 variants (c.382C>T; pR128X). Details of his case up to age 11.2 years have been reported previously[12]. Treatment with liothyronine (T3) improved linear growth and has been continued through adolescence into adulthood to maintain normal FT3 levels. He showed aortic root dilatation (3.8 cm) at age 14 years, which is progressing (16 years, 3.9 cm; 24 years, 4.3 cm) with effacement of the sinotubular junction (Fig. 1A). He is monitored echocardiographically annually but has not required surgery as yet, precluding analyses of aortic histology and VSMCs. He exhibits reduced serum selenoproteins, diminished selenoprotein synthesis in peripheral blood mononuclear cells together with reduced SECISBP2, selenoprotein mRNA and protein expression and thioredoxin reductase (TXRND) activity, and increased $H_2O_2$ production in dermal fibroblasts (Supplementary Fig. 3). Although muscle MR imaging (age 18 years) showed no changes, he is experiencing muscle weakness which limits exercise and prevents manual work. He has normal hearing, with no photosensitivity or Raynaud's disease.

Patient P4 (female, age 11 years) presented with delayed growth and short stature, muscle weakness and incoordination which limited some physical activities (riding a bike; jumping) at age 7 years. Investigations, showing raised circulating free T4, low free T3 and normal TSH concentrations prompted genetic testing, identifying a biallelic deletion in SECISBP2 (Chr9:91935388-91941101del), generating truncated SECISBP2 protein (pE61E*4) similar to the product of a variant allele in an unrelated previous case[8] (Supplementary Fig. 4A, B). As recorded in other cases[13], she responded well to growth hormone therapy, but diminishing dosage requirements prompted its discontinuation after two years. She has normal hearing, no photosensitivity or symptoms of Raynaud's disease. Echocardiography at age 10 years showed aortic dilatation (annulus 18 mm, Z score +0.64; root 26 mm, Z score +1.24; sinotubular junction 21 mm, Z score +1.2; tract 21 mm, Z score +0.61) and valvular insufficiency (Fig. 1A). She exhibits raised, circulating thyroxine with low triiodothyronine and selenium levels (Supplementary Fig. 4C), selenoprotein deficiency in serum and fibroblasts and raised cellular $H_2O_2$ and membrane lipid peroxidation (Supplementary Fig. 5). Ongoing surveillance with muscle MRI and aortic echocardiography has been instituted.

Whole exome sequencing of 32 known aortopathy genes (ABL1, ACTA2, BGN, CBS, COL3A1, COL5A1, COL5A2, EFEMP2, ELN, FBLN5, FBN1, FBN2, FKBP14, FLNA, FOXE3, LOX, MFAP5, MYH11, MYLK, NOTCH1, PLOD1, PRKG1, SKI, SLC2A10, SMAD2, SMAD3, SMAD4, SMAD6, TGFB2, TGFB3, TGFBR1, TGFBR2) in P1, P2, P3 and P4 identified no pathogenic variants.

## Human aortopathy: changes in vessel wall and aortic smooth muscle cells

Histology of aneurysmal aorta following surgery in P1 (P1a, age 23 years; P1b, age 35 years) and P2 showed loss of medial architecture, with disruption of elastic laminae, Alcian Blue-positive ECM-filled 'cysts', microcalcification and VSMC apoptosis (Fig. 1B), but no inflammatory cell (macrophage) infiltrate (Supplementary Fig. 11). P1 and P2 aneurysmal but not control aortas showed expression of 8-oxoguanine (a product of reactive oxygen species-mediated DNA damage) and phosphorylated histone H2AX (a marker of DNA damage) in medial VSMCs (Fig. 1C). VSMCs cultured from P1 and P2 aortas exhibited increased $H_2O_2$ production (Fig. 2A) and peroxidation of cell (Fig. 2B) and mitochondrial phospholipids (Fig. 2C, D) and apoptosis

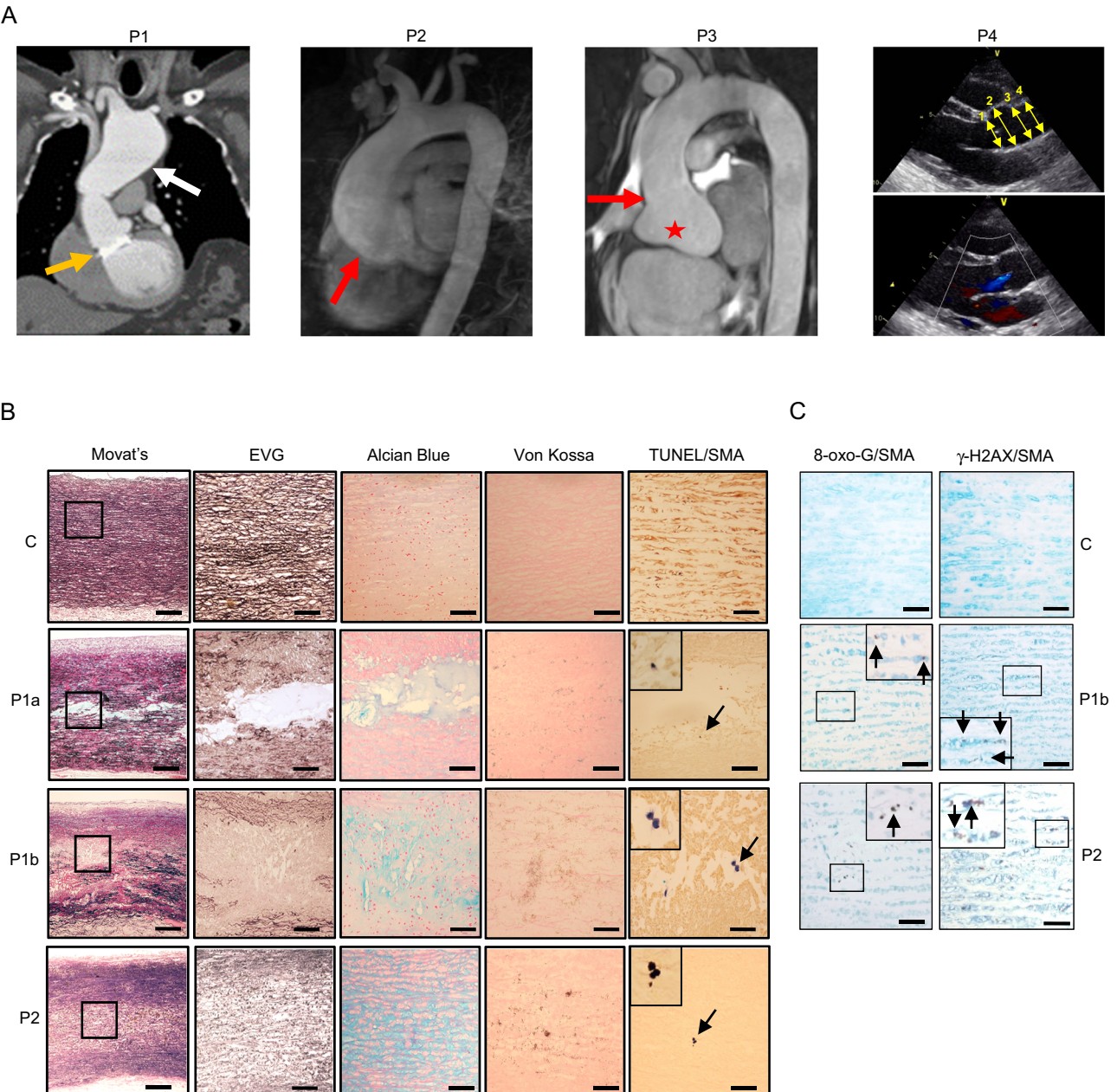

**Fig. 1 | Thoracic aortic aneurysms with cystic medial necrosis. A** CT imaging of P1 showing aortic valve and ascending aorta replacement at age 23 years (orange arrow) and distal arch dilatation (white arrow) at age 44 years. MR imaging of P2 and P3 showing dilatation of the aortic root (red star), with effacement of the sinotubular junction (red arrows). Echocardiogram of P4 showing positions (1, annulus; 2, root; 3, sinotubular junction; 4, tract) of abnormal aortic dimensions (upper panel) and aortic valvular insufficiency (lower panel). **B** Histochemistry and TUNEL/immunohistochemistry of control thoracic aorta (C), P1a (ascending aorta), P1b (aortic arch) or P2 (ascending aorta), using Movat's pentachrome, Elastin van Gieson (EVG), Alcian Blue, and von Kossa stains, and terminal deoxynucleotidyl transferase dUTP nick end labelling (TUNEL) with anti-α-smooth muscle actin (SMA/Acta2). Outlined areas in Movat's sections are shown in higher power in subsequent stains. Insets in TUNEL/SMA sections are high-power views of TUNEL-positive cells (arrows). Scale bars: 1 mm in Movat's, 250 µm in other panels. **C** Immunohistochemistry of control ascending aorta (C), or P1b and P2 thoracic aorta for 8-oxoguanine (8-Oxo-G, left panels) or γ-H2AX (right panels), co-stained with anti-SMA. Insets are high-power views of outlined areas. Scale bars: 250 µm. **B** and **C** $n = 1$ independent experiment.

(Fig. 2E), consistent with deficiency and reduced activity of antioxidant selenoenzymes (glutathione peroxidases, thioredoxin reductases) and other selenoproteins (Supplementary Fig. 2A–E) in these cells. In keeping with their lack of glutathione peroxidase type 4 (Supplementary Fig. 2B), which protects cells from iron-dependent cell death or ferroptosis[14], patient-derived aortic VSMCs exhibited increased oxidative damage and reduced cell viability following exposure to erastin (Fig. 2F, G), a small molecule activator of this cell death pathway[14]. Exposure of patient-derived aortic VSMCs to general (α-tocopherol) or mitochondria-targeted (MitoQ) chain-breaking

antioxidants or an iron chelator (desferrioxamine), reversed increased peroxidation of membrane lipids (Fig. 2B, F) and apoptosis (Fig. 2E) or reduced cell viability (Fig. 2G). Other antioxidants (Ebselen (a glutathione peroxidase/peroxiredoxin mimetic), N-acetylcysteine or ascorbate) were much less effective (Supplementary Fig. 6). Supplementing P2's dietary intake with vitamin E (α-tocopherol), protected lipids in membranes of his peripheral blood mononuclear cells or plasma from peroxidation (Fig. 2H, I), with unexpected improvement of his known, enhanced, systemic insulin sensitivity and favourable lipid profile (Table 1), previously attributed to elevated ROS[8].

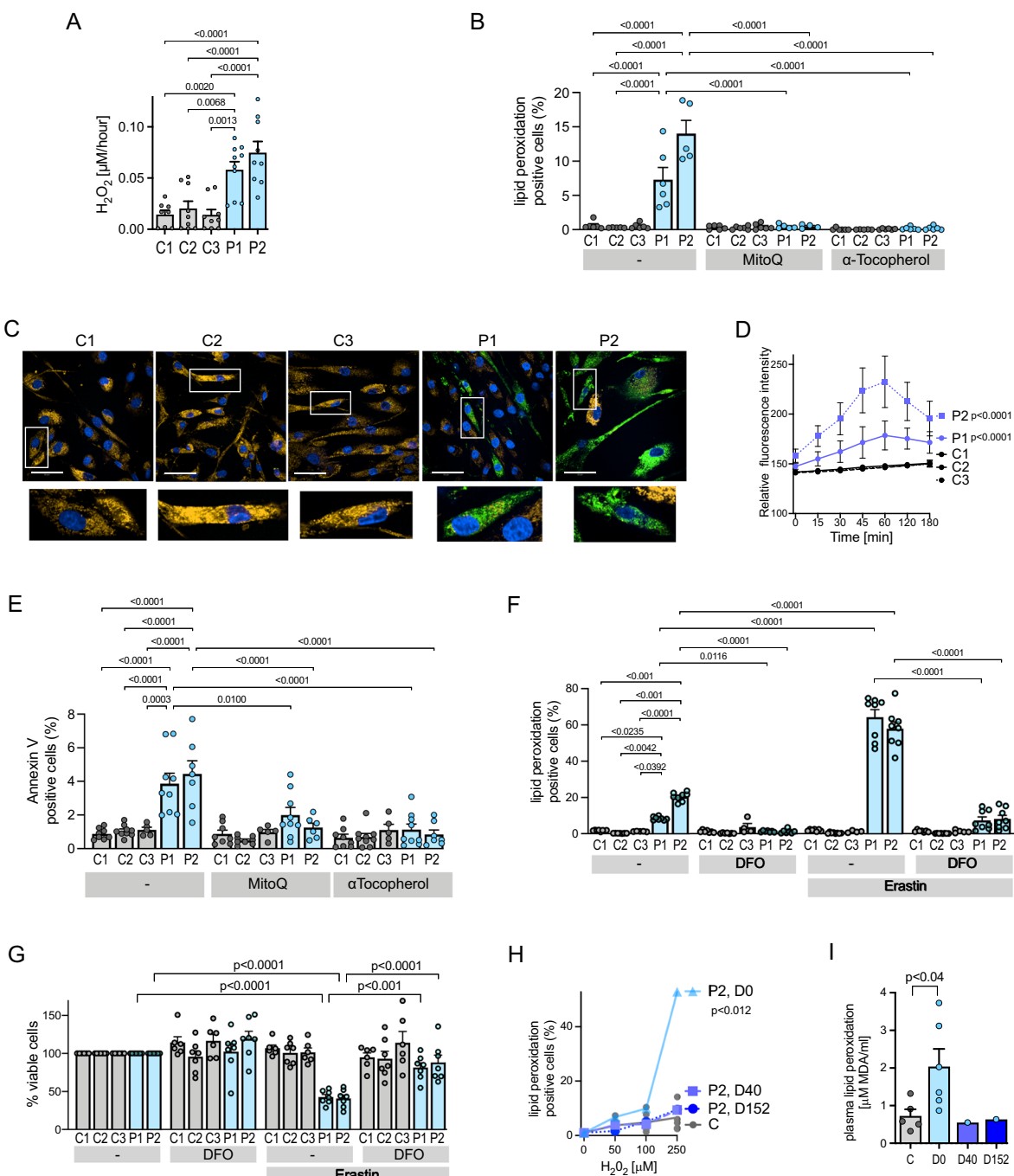

Consistent with preserved selenoprotein synthesis in heterozygous family members of our patients[8] (Supplementary Figs. 3, 5), 25 heterozygous carriers of rare, deleterious *SECISBP2* variants, among 35,000 UK Biobank participants in whom both cardiac MRI and whole exome sequencing data were available[15] (Supplementary Tables 8, 9), exhibited no significant differences in thoracic aortic diameter (Supplementary Table 1).

**Aortopathy in zebrafish with global Secisbp2 deficiency**

To investigate the role of *Secisbp2* in aneurysm formation, we studied *Secisbp2*$^{Q333X/Q333X}$ zebrafish[16] (Supplementary Fig. 7A, B) that show deficiency of Secisbp2 (Supplementary Fig. 7C, D) and most selenoproteins including antioxidant selenoenzymes (Supplementary Fig. 7E, F) and abnormal thyroid function (Supplementary Fig. 7G–J), similar to patients with *SECISBP2* mutations. The ventral

aorta (VA) and bulbus arteriosus (BA) were significantly dilated in homozygous mutant *Secisbp2*$^{Q333X/Q333X}$ adult zebrafish compared to wild-type or heterozygous counterparts (Fig. 3A–C). VA/BA tissue showed increased $H_2O_2$ content, lipid peroxidation and expression of the DNA damage and proapoptotic genes p53 and Casp3 (Fig. 3D–F). Exposure of *Secisbp2*$^{Q333X/Q333X}$ (but not wild type or heterozygous *Secisbp*$^{WT/Q333X}$) embryos to $H_2O_2$ or erastin caused aortic dilatation and significantly reduced their survival (Fig. 3G–O), with addition of antioxidants (α-tocopherol, MitoQ) or desferrioxamine preventing these deleterious effects (Fig. 3G–O).

In Secisbp2 morpholino-treated zebrafish (with blockade of Secisbp2 synthesis) (Supplementary Fig. 8A–E), aortic dilatation (Fig. 4A, D, E) was worsened by exposure to $H_2O_2$ or (Fig. 4G–I) or erastin (Fig. 4J–L), with increased embryo $H_2O_2$ content (Fig. 4B) oxidative damage (Fig. 4C) and apoptosis (Fig. 4F). These abnormalities

**Fig. 2 | Increased oxidative stress, apoptosis and susceptibility to ferroptosis in aortic vascular smooth muscle cells. A** $H_2O_2$ production in aortic VSMCs from patients P1, P2 (blue bars) and controls (C1, C2, C3: grey bars). Statistics: Ordinary one-way ANOVA with adjusted $P$ values (Tukey's multiple comparison test), each bar represents the mean ($n \geq 8$ independent experiments), error bars represent SEM, p-values compare P1 and P2 with controls. **B** Membrane lipid peroxidation in aortic VSMCs from patients P1, P2 (blue bars) and controls (C1–C3: grey bars), with exposure to the vehicle, MitoQ or α-tocopherol. Statistics: Two-way ANOVA with adjusted $P$ values (Tukey's multiple comparison test), each bar represents the mean ($n \geq 4$ independent experiments), error bars represent SEM, p-values compare P1 and P2 with controls, or vehicle-treated data from P1 and P2 with antioxidant-treated cells. **C** Representative microscopic images of primary, aortic vascular smooth muscle cells from controls (C1, C3, C3) or patients (P1, P2) following 1 h exposure to MitoPerOx, a fluorescent probe assessing lipid peroxidation within mitochondria, with insets below showing high power views of single cells. Each image was acquired independently ten times, with similar results. Scale bars: 50 μm. **D** Quantitation of relative fluorescence intensity (RFI) in primary VSMCs from patients (blue lines; P1 solid line; P2 broken line) and three controls (C1, C3, C3: black lines), treated with MitoPerOx over the time period indicated. Statistics: Unpaired, two-tailed $t$-test, error bars represent SEM, p-values compare patient versus controls ($n = 10$ independent experiments). **E** Apoptosis (Annexin V positive

cells) of aortic VSMCs from patients P1, P2 (blue bars) and controls (C1–C3: grey bars), with exposure to the vehicle, MitoQ or α-tocopherol. Statistics: Two-way ANOVA with adjusted $p$ values (Tukey's multiple comparison test), each bar represents the mean ($n \geq 5$ independent experiments), error bars represent SEM, p-values compare P1 and P2 with controls, or vehicle-treated data from P1 and P2 with antioxidant-treated cells. **F** and **G.** Membrane lipid peroxidation (**F**) and cell viability (**G**) of aortic VSMCs from patients (P1, P2: blue bars) and controls (C1–C3: grey bars), with exposure to vehicle, erastin or desferrioxamine (DFO). Statistics: Two-way ANOVA with adjusted $p$ values (Tukey's multiple comparison test), each bar represents the mean (**F:** $n \geq 4$; **G:** $n = 6$ independent experiments), error bars represent SEM, p-values in **F** compare untreated/treated P1 and P2 with control cells treated similarly (ns: not significant). **H** and **I** $H_2O_2$-induced peroxidation of membrane lipid in PBMCs (**J**) or serum (**K**) from patient P2 at different days of dietary supplementation with α-tocopherol (D0: day 0, light blue), 40 (D40: day 40, shaded blue) or 152 (D152: day 152, dark blue) or untreated control subjects (C: grey). Statistics: **H** Two-way ANOVA, each line represents the mean for controls ($n = 5$) and single values for P2, $p$-value compares P2 D0 with controls. **I** Unpaired, two-tailed $t$-test, each bar represents the mean for controls ($n = 5$) and untreated P2 ($n = 6$), or single values for P2 treated for different numbers of days as indicated. Source data are provided as a Source Data file.

were rescued by either coexpressing Secisbp2 mRNA (Fig. 4A–F) or addition of antioxidants (α-tocopherol, MitoQ) (Fig. 4G–I) or desferrioxamine (Fig. 4J–L).

## Aortopathy in mice with vascular smooth muscle cell Secisbp2 deficiency

To examine the effect of Secisbp2 deficiency in VSMCs, we generated Myh11-Cre[ERt2]/*Secisbp2*[flox/flox] mice and treated 6–8-week-old animals with tamoxifen to conditionally abolish *Secisbp2* gene and selenoprotein expression selectively in the arterial media (Fig. 5A, B; Supplementary Figs. 9, 10). Subsequent infusion of angiotensin II, an established model of thoracic and abdominal aortic aneurysm (AAA) formation[17], caused markedly diminished survival of homozygous

mice compared to wild-type and heterozygous littermates (Fig. 5C), associated with significantly increased TAA development (Fig. 5D; Supplementary Fig. 9C), dissection (Fig. 5E) and rupture. Histology of the TAAs, before dissection or rupture, showed loss of medial architecture, with disruption of elastic laminae, Alcian Blue-positive ECM-filled 'cysts', and microcalcification (typical features of cystic medial necrosis due to VSMC apoptosis in mice[18] (Fig. 5F). Macrophages were present at similar levels in TAAs from both wild type and VSMC-specific *Secisbp2* knockout mice treated with angiotensin II (Supplementary Fig. 11), consistent with this cellular infiltrate being a recognized feature of this animal model[19]. TAAs in homozygous animals showed increased VSMCs expressing 8-oxo-guanine and γ-H2AX and apoptosis (Fig. 5F–I) compared with wild-type and heterozygous littermates.

## Discussion

We have identified early onset (age 10–37 years), progressive, dilatation of the ascending aorta in four patients with a multisystem disorder due to mutations in *SECISBP2*, resulting in deficiencies of selenocysteine-containing proteins. The cystic medial degeneration in our patients resembles the canonical histology of early-onset aneurysms in other multisystem syndromes (e.g., Marfan, Loeys-Dietz, Ehlers-Danlos Type IV) with a differing genetic basis. Aortic dilatation has not yet been documented in nine other known cases of selenoprotein deficiency, but as other patients harbour different, pathogenic *SECISBP2* variants or were described in childhood or early adolescence (age 2–14 years)[20], it is conceivable that the development of aortopathy may vary, either depending on underlying *SECISBP2* variant genotype or manifest at an older age due to oxidative tissue damage being cumulative.

We used two animal models to determine a causal relationship between *Secisbp2* deficiency and TAA development. Homozygous *Secisbp2*-null adult zebrafish showed ventral aortic dilatation, associated with marked upregulation of markers of oxidative stress, DNA damage and apoptosis, with Secisbp2-null embryos exhibiting aortic dilatation and diminished survival either following exposure to $H_2O_2$ or activation of the ferroptosis pathway. Prevention of these abnormalities by exposure to antioxidants or desferrioxamine, strongly suggests that the aortic phenotype is mediated by increased oxidative stress and possibly susceptibility to ferroptotic cell death, secondary to deficiency of antioxidant selenoenzymes, including GPX4.

Angiotensin II infusion induces oxidative stress, DNA damage, and VSMC death in mice[21–23] and also induces TAA aneurysm formation and

**Table 1 | Circulating biochemical measurements in patient P2 before and after dietary vitamin E (α-tocopherol) supplementation**

| P2 | | Day 0 | Day 40 | Day 152 |
|---|---|---|---|---|
| Treatment | α-tocopherol (mg) | NONE | 150 | 150 |
| | Vitamin D (IU) | 400 | 400 | 400 |
| | Pregabalin (mg) | 100 | 100 | 100 |
| | Bisoprolol (mg) | 2.5 | 2.5 | 2.5 |
| | Losartan (mg) | 25 | 25 | 25 |
| | Spironolactone (mg) | 25 | 25 | 25 |
| Vitamin E (15–45 µmol/L) | | 24 | 48 | 37 |
| Vitamin E/Cholesterol ratio (3.5–9.5 µmol/mmol cholesterol) | | 5.2 | 9.8 | 9.3 |
| TSH (0.5–5.50 mU/L) | | 1.72 | 1.88 | 1.71 |
| Free T4 (10.5–21.0 pmol/L) | | 39.5 | 35.6 | 33.7 |
| Free T3 (3.5–6.5 pmol/L) | | 3.5 | 3.7 | 3.1 |
| Reverse T3 (0.12–0.36 nmol/L) | | 1.07 | 1.15 | 1.11 |
| Selenium (0.90–1.70 µmol/L) | | 0.21 | 0.25 | 0.28 |
| Fasting Glucose (3.5–6.0 mmol/L) | | 4.7 | 4.4 | 4.6 |
| Insulin (0–11 mU/L) | | 12.1 | 6.3 | 4.6 |
| HOMA-IR (Optimal < 1.0) | | 2.9 | 1.4 | 1.1 |
| Adiponectin (0.5–30 µg/mL) | | 41.5 | 41.4 | 51.8 |
| LDL cholesterol (<3.0 mmol/L) | | 3.35 | 3.19 | 2.69 |
| HDL cholesterol (>1.03 mmol/L) | | 1.16 | 1.16 | 1.05 |
| Triglyceride (<1.7 mmol/L) | | 1.08 | 1.43 | 1.0 |

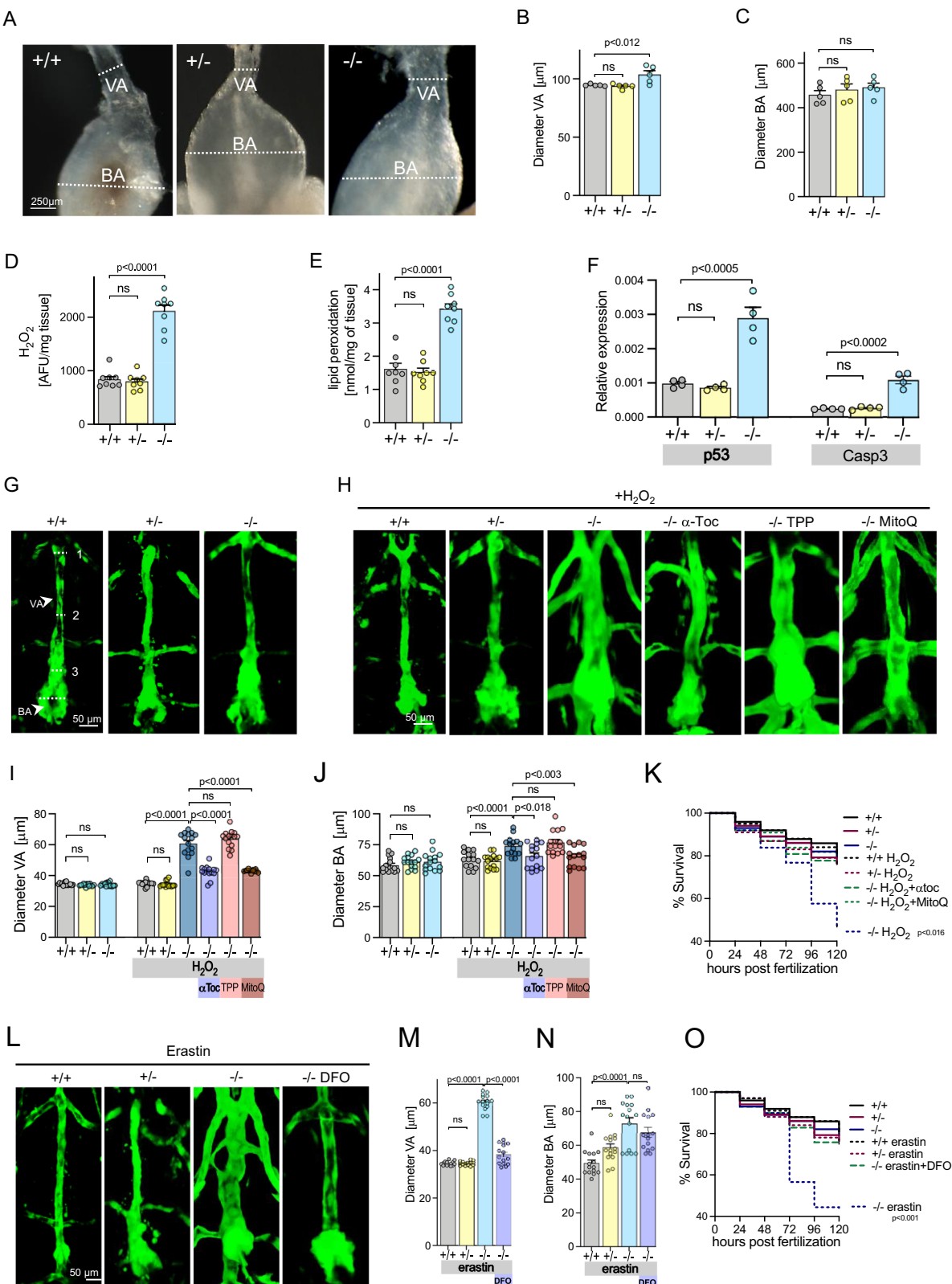

dissection[17]. We observed similar survival, aneurysm formation, oxidative damage and VSMC apoptosis in thoracic aortae of wild-type and heterozygous, VSMC-specific *Secisbp2* knockout mice, suggesting that the product of a single *Secisbp2* allele mediates sufficient selenoprotein synthesis to prevent TAAs. In contrast, homozygous, VSMC-specific *Secisbp2* knockout mice had an increased frequency of TAAs, with greatly increased oxidative DNA damage and VSMC apoptosis and

histological features similar to those seen in our patients. Aortopathy occurring in VSMC-specific *Secisbp2* null animals suggests that VSMCs are the cell type that underlies TAA formation due to the loss of SECISBP2 activity.

Loss of antioxidant selenoenzymes, increased hydrogen peroxide, cell membrane and mitochondrial phospholipid peroxidation, oxidative DNA damage, apoptosis and susceptibility to ferroptosis were

**Fig. 3 | Aortic abnormalities in *Secisbp2* mutant zebrafish. A** Representative aortic outflow tract images, comprising ventral aorta (VA) and bulbus arteriosus (BA), from wild-type (Secisbp2$^{wt/wt}$; +/+), heterozygous (Secisbp2$^{Q333X/wt}$; +/−) and homozygous (Secisbp2$^{Q333X/Q333X}$; −/−) mutant adult (age 6 months) zebrafish. The dotted lines show the positions of VA and BA measurements quantified in (**B** and **C**). Each image was repeated independently 5 times with similar results. Scale bar: 250 μm. **B** and **C** VA and BA diameters, adjusted for body size, measured at dotted line positions in (**A**), in wild-type (+/+: grey bar), heterozygous (+/−: yellow bar) and homozygous (−/−: blue bar) *Secisbp2* mutant fish. Statistics: Unpaired, two-tailed *t*-test, each bar represents the mean (*n* = 5), and error bars represent SD. **D−F** H$_2$O$_2$ (**D**, *n* = 8), lipid peroxidation (**E**, *n* = 8) and p53/casp3 mRNA expression (**F**, *n* = 4) in VA/BA tissue from wild type (+/+: grey bar), heterozygous (+/−: yellow bar) and homozygous (−/−: blue bar) *Secisbp2* mutant fish. Statistics: unpaired, two-tailed *t*-test, each bar represents the mean, and error bars represent SD. **G**, **H** and **L**. Representative images, of the ventral aorta (VA) and bulbus arteriosus (BA) from wild-type (Secisbp2$^{wt/wt}$; +/+), heterozygous (Secisbp2$^{Q333X/wt}$; +/−) and homozygous (Secisbp2$^{Q333X/Q333X}$; −/−) zebrafish embryos at 5 dpf. Images show vascular endothelial cell-specific green fluorescent protein, VA diameter and BA diameter was measured at positions (1–3) marked with broken line (**G**, +/+). Zygotes were treated (from 3 to 120 hpf) with 0.5 mM H$_2$O$_2$ or H$_2$O$_2$ + 1 mM α-tocopherol (αToc) or H$_2$O$_2$ + 1 mM Decyltriphenylphosphonium (TPP) or H$_2$O$_2$ + 1 mM MitoQ (**H**) or treated (from 6 to 120 hpf) with 10 mM Erastin or 10 mM Erastin + 100 mM desferrioxamine (DFO) (**L**). Each image was acquired independently fifteen times, with similar results. Scale bars: 250 μm. **I** and **J** Mean VA diameter, measured at positions (1–3) marked in (**A**) (**I**), or BA diameter (**J**) quantified in different groups (5 dpf, *n* = 15) in wild-type (Secisbp2$^{wt/wt}$; +/+), heterozygous (Secisbp2$^{Q333X/wt}$; +/−), homozygous (Secisbp2$^{Q333X/Q333X}$; −/−) fish, untreated or treated with 0.5 mM H$_2$O$_2$ in combination with 1 mM α-tocopherol (αToc) or 1 mM Decyltriphenylphosphonium (TPP) or 1 mM MitoQ. Statistics: unpaired, two-tailed *t*-test, each bar represents the mean and error bars represent SD. **K** Percent survival of embryos over time in different groups described in (**G–J**). Statistics: Logrank test for trend (Chi-square), *p*-value for H$_2$O$_2$ treated Secisbp2$^{Q333X/Q333X}$ (−/−) versus all comparisons, each line represents the % of surviving fish (*n* = 100−200). **M** and **N** Mean VA diameter, measured at positions (1–3) marked in (**A**) (**M**), or BA diameter (**N**) quantified in different groups (5 dpf, *n* = 15) in wild-type (Secisbp2$^{wt/wt}$; +/+), heterozygous (Secisbp2$^{Q333X/wt}$; +/−), homozygous (Secisbp2$^{Q333X/Q333X}$; −/−) fish, all treated with 10 mM Erastin or with 10 mM Erastin + 100 mM desferrioxamine (DFO). Statistics: unpaired, two-tailed *t*-test, each bar represents the mean, error bars represent SD. **O** Percent survival of embryos over time in different groups described in (**L–I**). Statistics: Logrank test for trend (Chi-square), the *p*-value for elastin-treated Secisbp2$^{Q333X/Q333X}$ (−/−) versus all comparisons, each line represents the percentage of surviving fish (*n* = 100−200). Source data are provided as a Source Data file.

features common to aortic cells or tissue from patients or zebrafish with biallelic *SECISBP2/secisbp2* defects, with increased oxidative damage and cell death also evident in a mouse model of Secisbp2 inactivation. Furthermore, exposure of aortic VSMCs from patients or *Secisbp2* mutant embryos to α-tocopherol (vitamin E) or MitoQ, which inhibit lipid peroxidation either throughout the cell or within mitochondria respectively, markedly attenuated both lipid peroxidation and apoptosis, directly implicating excess ROS in the pathogenesis of these processes.

Our findings accord with previous observations that implicate oxidative stress and DNA damage in pathogenesis of ascending aortic aneurysm[24]. Increased H$_2$O$_2$ production is seen in aneurysm VSMCs from patients with Marfan syndrome[25], and TAA tissue from Loeys-Dietz syndrome patients exhibits diminished activity of antioxidant enzymes (GST, GPX, TXNRD) together with increased lipid peroxidation[26]. Murine models of Marfan (Fibrillin-1 mutant), smooth muscle α-actin deficiency and syndromic aortopathy (Fibulin 4 mutant) also show increased oxidative stress and ROS production[25,27,28]. Furthermore, NADPH oxidase, a major enzymatic source of ROS, is upregulated in human AAA tissue, and inhibition of this pathway attenuates aneurysm formation in animal models[29].

Whilst we contend that deficiency of antioxidant selenoenzymes is central to the pathogenesis of aortopathy in patients with deleterious *SECISBP2* variants, we do not exclude a role for other selenoproteins. For example, expression of selenoprotein S, which prevents apoptosis of rodent VSMCs[30], was reduced in aortic VSMCs of P1 and P2 patients (Supplementary Fig. 2). We also do not discount the possibility that selenoprotein deficiencies in our patients perturb signalling pathways known to contribute to aortopathy[2]. Prevention of oxidative damage and apoptosis in patient-derived aortic VSMCs or aortic dilatation in the zebrafish model by antioxidants, together with proven efficacies of α-tocopherol or a mitochondrial ROS scavenger in preventing AAA formation in mouse models[31], raises the possibility of antioxidant therapy to prevent aortopathy in patients with *SECISBP2* mutations. Our observation that dietary vitamin E (α-tocopherol) supplementation in patient P2 can prevent oxidative damage without adverse effects on beneficial phenotypes (e.g., systemic insulin sensitivity, favourable lipid profile), provides proof-of-concept for future trialling of antioxidant and/or iron-chelating therapies in this disorder.

Finally, as not all genetic aetiologies of TAA and dissection are known[3], we suggest that *SECISBP2* should be included in gene panels for TAA, particularly in cases of idiopathic aortopathy with abnormal circulating thyroid hormone and plasma selenium levels.

## Methods

### Genetic and phenotypic analyses

All studies in patients and control subjects, including analyses of participant-derived cells and tissues, were approved by Local Research Ethics Committees (Cambridgeshire Local Research Ethics Committee, LREC 98/154; Medical Ethics Committee, Erasmus MC, MEC-2015-362) and undertaken with prior written, informed consent of participants and/or parents. The authors affirm that human research participants or their parents/legal guardians provided written informed consent for the publication of the potentially identifiable medical data included in this article.

### Biochemical measurements

Serum T4, FT4, T3 and TSH were measured by chemiluminescence assays (Vitros ECI Immunodiagnostic System; Ortho-Clinical Diagnostics Inc., Rochester, NY, USA) and rT3 by a commercial RIA (Immunodiagnostic Systems, Scottsdale, AZ, USA). Sex hormone binding globulin (SHBG) was measured using an immunometric method (Immulite 2000). Plasma selenium was measured by inductively coupled plasma mass spectrometry (ICPMS; Thermo Elemental). Plasma vitamin E was measured by ultra-performance liquid chromatography (Scottish Trace Element and Micronutrient Diagnostic Research Laboratory, Glasgow, UK). Serum lipid peroxides were measured as thiobarbituric acid reactive substances (TBARS) using a method described by the manufacturer (R&D systems, Abingdon, UK). Circulating glucose and lipids were measured by immunoassay (Siemens Attelica, Erlangen, Germany), with insulin and adiponectin by two-step, time-resolved immunofluorometric (AutoDELFIA, Perkin-Elmer, UK) assay.

### Whole-exome sequencing studies

Genomic DNA from whole blood (P1, P4), was processed (Illumina TruSeq DNA Library preparation, Illumina Inc., San Diego, CA, USA), exome captured (Nimblegen SeqCap EZ, Roche Nimblegen, Inc., Madison, WI, USA) and paired-end sequenced (Illumina HiSeq2000 sequencer with TruSeq V3 chemistry). Reads were aligned to the human reference genome hg19 (UCSC, Genome Reference Consortium GRCh37), DNA variants called (UnifiedGenotyper, GATK v2.7.4) and annotated (NCBI Reference Sequence Database) using ANNOVAR (version 2014-07-14)[32]. Heterozygous variants in *SECISBP2* (c.1312A>T; p.K438X; c.18947-3C>G p.D631Efs*4) were validated by Sanger sequencing (Supplementary Fig. 1 and Supplementary Table 2). Similarly, genomic DNA from P1–P4 was sequenced (Twist

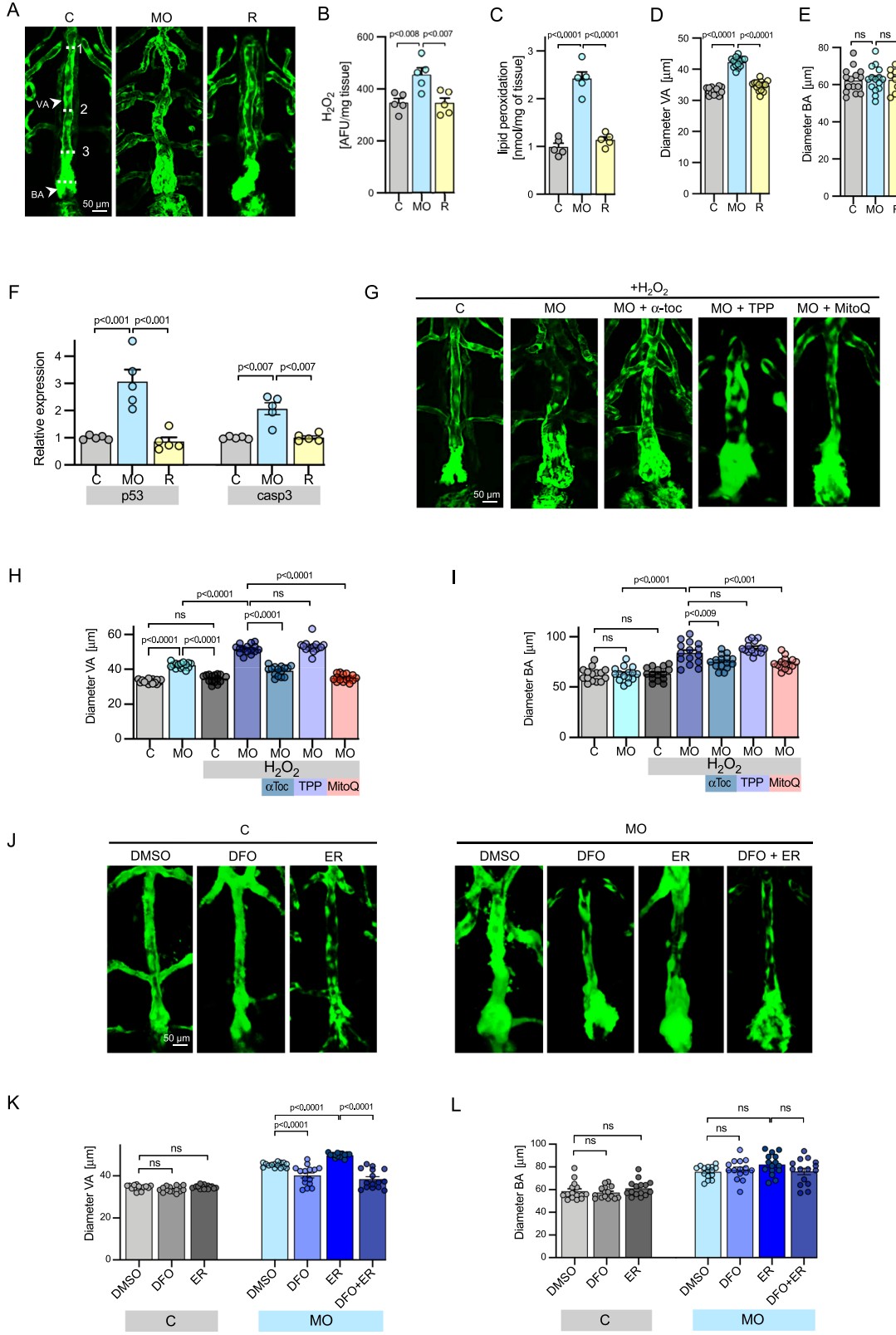

comprehensive Exome on Illumina NovaSeq platform, Cambridge Genomics Laboratory, Cambridge University Hospitals, UK), with analyses of candidate aortopathy genes (*ABL1, ACTA2, BGN, CBS, COL3A1, COL5A1, COL5A2, EFEMP2, ELN, FBLN5, FBN1, FBN2, FKBP14, FLNA, FOXE3, LOX, MFAP5, MYH11, MYLK, NOTCH1, PLOD1, PRKG1, SKI, SLC2A10, SMAD2, SMAD3, SMAD4, SMAD6, TGFB2, TGFB3, TGFBR1, TGFBR2).*

## Aortic traits in carriers of heterozygous, loss-of-function, SECISBP2 variants in UK Biobank

The UK Biobank (UKB) is a population-based prospective study from the United Kingdom with deep phenotypic data on up to 500,000 individuals[33], including cardiac MRI for over 43,000 participants[15,34], and exome sequencing on over 450,000 participants[35]. The UK Biobank resource was approved by the UK Biobank Research Ethics

**Fig. 4 | Aortic abnormalities in *Secisbp2* knockdown zebrafish. A, G** and **J** Representative aortic images, visualized by green fluorescent protein expression as described in Fig. 3 legend, of ventral aorta (VA) and bulbus arteriosus (BA) of zebrafish embryos (5dpf), following zygote injection with control (C), or *Secisbp2* morpholinos (MO) or co-injection of *Secisbp2* morpholino with *Secisbp2* mRNA (Rescue; R). Zygotes were treated (3–120 hpf) with 0.5 mM $H_2O_2$ or $H_2O_2$ + 1 mM a-tocopherol (aToc) or $H_2O_2$ + 1 mM Decyltriphenylphosphonium (TPP) or $H_2O_2$ + 1 mM MitoQ (**G**) or treated (from 6 to 120 hpf) with vehicle (DMSO), 100 mM desferrioxiamine (DFO), 10 mM Erastin or 10 mM Erastin + 100 mM DFO (**J**). The dotted lines **A** show the positions of VA and BA measurements quantified in (**B** and **C**). Each image was repeated independently 15 times with similar results. Scale bar: 50 μm. **B, C** and **F** $H_2O_2$ production (**B**), membrane lipid peroxidation (**C**) and p53 and casp3 mRNA expression (**F**) in VA of 5 pools of 50 embryos at 4 dpf following

zygote injection with control (C: grey), or *Secisbp2* morpholinos (MO: blue) or co-injection of *Secisbp2* morpholino with *Secisbp2* mRNA (Rescue; R: yellow). Statistics: One-way ANOVA with adjusted *p* values (Tukey's multiple comparison test), each bar represents the mean of *n* = 5 independent experiments, and error bars represent SEM. **D, E, H, I, K** and **L**. VA diameter, measured at positions (1–3) marked in A (**D, H, K**) or BA diameter (**E, I, L, L**) quantified in different groups (5 dpf, *n* = 15 independent experiments) following zygote injection with control (C), or *Secisbp2* morpholinos (MO) or co-injection of *Secisbp2* morpholino with *Secisbp2* mRNA (Rescue; R) and treated as indicated (0.5 mM $H_2O_2$; 1 mM α-tocopherol (αToc); 1 mM Decyltriphenylphosphonium (TPP); 1 mM MitoQ). Statistics: Ordinary one-way ANOVA with adjusted *p* values (Tukey's multiple comparison test), each bar represents the mean, error bars represent SEM, ns = not significant. Source data are provided as a Source Data file.

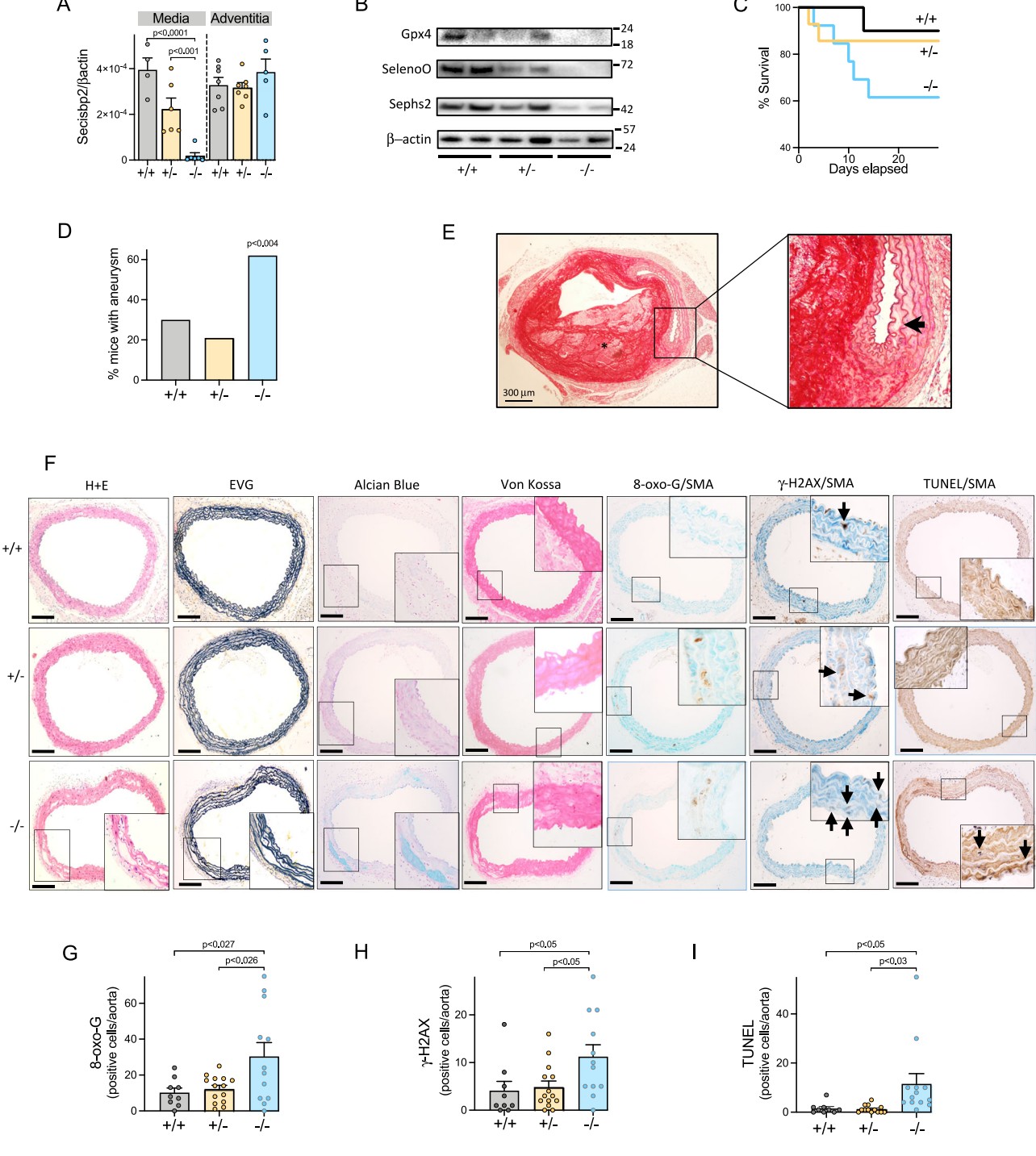

**Fig. 5 | Thoracic aortic aneurysms in VSMC-targeted, *Secisbp2* knockout mice.** **A** *Secisbp2* mRNA expression in aortic media or adventitia tissue isolated from wild-type (+/+: grey), heterozygous (+/−: orange) and homozygous (−/−: blue) VSMC-specific *Secisbp2* knockout mice. Statistics: ordinary one-way ANOVA with adjusted *p* values (Tukey's multiple comparison test), each bar represents the mean (*n* = 4−7 animals per genotype), error bars represent SEM. **B** Expression of selenoproteins (Gpx4, SelenoO, Sephs2) or β-actin (loading control) in aortic media tissue isolated from wild-type (+/+), heterozygous (+/−) and homozygous (−/−) VSMC-specific *Secisbp2* knockout mice (*n* = 2 animals per genotype). **C** and **D** Percent survival of animals (**C**) or aortic aneurysm frequency (**D**), following angiotensin II infusion of wild type (+/+: grey), heterozygous (+/−: orange) or homozygous (−/−: blue) VSMC-specific, *Secisbp2* knockout mice. Statistics: Two sample *Z* test of proportions, two-tail for −/− vs. +/− plus +/+ group comparison (*n* = 10−14 animals per genotype). **E** Histology of haematoxylin and eosin (H&E)-stained TAA (from angiotensin II-treated homozygous VSMC-specific *Secisbp2* knockout mouse showing laminated thrombus within the vessel wall (asterisk) and cystic medial necrosis (arrowed inset). *n* = 10 independent experiments with similar results, scale bars = 300 μm. **F** Histology of thoracic aorta stained with H&E, EVG, Alcian Blue and von Kossa stains, or immunohistochemistry for 8-oxo-G, γ-H2AX or TUNEL with SMA from wild-type (+/+), heterozygous (+/−) and homozygous (−/−) VSMC-specific *Secisbp2* knockout mice exposed to angiotensin II (as detailed in Supplementary Fig. 9B). Insets show high power views of outlined areas. Arrows indicate γ-H2AX or TUNEL-positive, αSMA-expressing cells, *n* = 9−14 independent experiments with similar results, scale bars = 300 μm. **G−I** Mean number of 8-oxo-G (**G**), γ-H2AX (**H**) and TUNEL (**I**)-positive cells in thoracic aorta sections of wild-type (+/+: grey), heterozygous (+/−: orange) and homozygous (−/−: blue) VSMC-specific *Secisbp2* knockout mice. *$p < 0.05$ 1-Way ANOVA. *n* = 9−14 mice 5 sections/mouse. Statistics: ordinary one-way ANOVA with adjusted *p* values (Tukey's multiple comparison test), each bar represents the mean (*n* = 9−14 mice, 5 sections/mouse), error bars represent SEM. The genetic background of the Myh11-Cre^ERt2/Secisbp2^flox/flox mice generates only male, VSMC-specific knockout animals (C57BL/6N, 13 weeks old). Source data are provided as a Source Data file.

Committee and all participants provided written informed consent to participate. Use of UK Biobank data was performed under application numbers 7089 and 17488. The application protocol was approved by the local Massachusetts General Hospital Institutional Review Board.

We sought potential aortic phenotypes in heterozygous carriers of loss-of-function (LOF) *SECISBP2* variants in UKB. We previously trained and applied a deep learning model to measure the diameters of the ascending and descending thoracic aorta during the cardiac cycle for UK Biobank participants from their MRI data[15]. In the present study, we utilized both the raw aortic diameter measurements (ascending diameter in systole, ascending diameter in diastole, descending diameter in systole, descending diameter in diastole)[15], as well as inverse-rank normalized (IRN) values of these traits.

To identify carriers of *SECISBP2* variants we utilized the exome sequencing data, generated as described elsewhere (https://www.ukbiobank.ac.uk/media/najcnoaz/access_064-uk-biobank-exome-release-faq_v11-1_final-002.pdf; https://biobank.ndph.ox.ac.uk/showcase/ukb/docs/UKB_WES_Protocol.pdf). We used the original quality functionally equivalent (OQFE) exome call set and undertook stringent quality control (QC) of the exome sequencing data using a previous pipeline[36]. High-quality exome sequencing data and aortic MRI measurements were available for between 33,618 and 34,618 participants.

Variants were annotated using the Loss-of-Function Transcript Effect Estimator (LOFTEE[37]) plug-in implemented in the Variant Effect Predictor (VEP; v.105)[38] (https://github.com/konradjk/loftee). LOFTEE was implemented to identify high-confidence LOF variants for the canonical transcript of *SECISBP2*, which include frameshift indels, stop-gain variants and splice site disrupting variants. We further annotated variants with continental allele frequencies from gnomAD[37], identifying minor allele frequency (MAF) thresholds (MAF < 0.1% in exome and gnomAD datasets) for inclusion of LOF variants. After annotation, we identified 25 carriers of rare (MAF < 0.1%) *SECISBP2* LOF variants amongst the participants.

We then performed gene-based burden testing for the two *SECISBP2* LOF masks across each of the included aortic phenotypes, using linear, whole-genome, ridge regression models implemented in REGENIE[39], which have been shown to produce similar results to mixed-effects models, generally handling population stratification and relatedness in biobank-scale datasets[39]. Additional fixed-effects covariates included: inferred sex, enrollment age, enrollment age squared, MRI time after enrollment (in years), MRI time after enrollment squared, MRI serial number, genotyping array, sequencing batch, and ancestral principal components 1−20.

### Histological and immunohistochemical (IHC) studies of human and murine aorta

Human and murine aortas were formaldehyde-fixed, dehydrated in graded ethanol, and embedded in paraffin. Aortic sections were analysed using standard histological stains as follows.

**Harris' haematoxylin and eosin (H&E).** Sections were placed in haematoxylin solution for 5 min. After several washes (destain solution: 250 mL 50% methanol, 250 mL Elix-10, 5 mL HCl) the slides were subsequently incubated for 5 min in Scott's solution (20 mg magnesium sulfate, 2 mg sodium bicarbonate in 1 l Elix-10), 6 min in eosin solution and rinsed with Elix-10. Sections were rapidly dehydrated, cleared in histoclear/xylene and mounted with Omnimount/DPX.

**Elastin Van Gieson (EVG).** Sections were placed in Verhoeff's haematoxylin (20 mL alcoholic haematoxylin, 8 mL 10% ferric chloride and 8 mL of Lugol's Iodine) for 30 min, immersed 10−15 s in differentiating solution (10 mL 10% ferric chloride mixed with 40 mL ddH$_2$O), 30 s in 5% hypo solution (5 g sodium thiosulfate in 100 mL distilled water) and counterstained in Van Gieson's solution for 5 min. Between the solutions, the slides were briefly rinsed in water. Sections were rapidly dehydrated, cleared in xylene and mounted with DPX.

**von Kossa.** Sections were incubated in 2% silver nitrate aqueous solution for 1 h under bright light, immersed in 3% aqueous sodium thiosulphate solution for 3 min and counterstained with 1% Neutral Red aqueous solution for 3−4 min. Between the solutions, the slides were briefly rinsed in water. Sections were rapidly dehydrated, cleared in xylene and mounted with DPX.

**Alcian Blue.** Sections were incubated in acetic acid solution for 3 min and then immersed in Alcian Blue (pH 2.5) for 30 min at room temperature. The slides were briefly rinsed in acetic acid solution and stained in Safranin O solution for 5 min, separated by rinsing for 2 min under running tap water. Sections were rapidly dehydrated, cleared in xylene and mounted with DPX.

**Movat's pentachrome.** The sections were stained in Elastic Stain solution (30 mL 5% haematoxylin, 15 mL 10% ferric chloride and 15 mL of Lugol's Iodine) for 20 min, dipped (15−20 times) in 2% ferric chloride differentiation solution, immersed in a 5% sodium thiosulfate solution for 1 min, incubated in a 3% acetic acid solution for 2 min, immersed in Alcian Blue (pH 2.5) for 25 min at room temperature (RT), all separated by a rinse in running tap water and a brief wash in distilled water. The slides were placed in Biebrich scarlet-acid fuchsin solution for 4 min, rinsed in running tap water and briefly washed in ddH$_2$O. The sections were differentiated in two changes of 5% phosphotungstic acid solution for 3 min each and briefly washed in distilled water. The slides were dipped 3−5 times in a 1% acetic acid solution, incubated in tartrazine solution for 3 min, rinsed in absolute alcohol, cleared and mounted with DPX.

**8-oxo-G/αSMA, γ-H2AX/SMA and CD68/MAC-3 IHC.** Slides were immersed in Dako Antigen Retrieval Solution pH 9 heated at 95−97 °C

for 30 min and left to cool for 20 min. After two 5 min washes in PBS, slides were protein-blocked for 10 min, and washed 1x in PBS. Slides were incubated with anti-8-oxo-G (Jaica N45.1, 1:3000), anti-γ-H2AX (Abcam 26350, 1:800), anti-CD68 (Dako M0814, 1:100) or anti-MAC-3 (BD Pharmingen 553322, 1:300) antibodies overnight at 4 °C. After a PBS wash, slides were incubated in Biotinylated Goat Anti-polyvalent antibody for 10 min at room temp, washed 4×, and incubated for 10 min with Streptavidin Peroxidase, washed 4×, and staining visualized with DAB. For double IHC, sections were then blocked with 20% BSA/PBS for 10 min @RT and incubated subsequently with anti-α-SMA (Dako M0851, 1:400) in 10% BSA for 1 h at RT, horse anti-mouse secondary antibody (Vector BA-1400; 1:400) for 30 min RT, Vector ABC-Alkaline Phosphatase (SK-5000) solution for 30 min @RT and Vector Alkaline Phosphatase Substrate III. (SK-5300), each separated by two washes with PBS for 5 min each.

**TUNEL.** Sections were incubated with 50 mg/mL Proteinase K (3–5 min/RT; pH 7.5), washed in ddH$_2$O, immersed in TdT buffer (Roche Diagnostics) for 5 min/RT and incubated with TdT enzyme (0.05–0.2 U/μL) + Digoxigenin-dUTP in humid atmosphere @ 37 °C for 30 min. The reaction was terminated by transferring slides to TB buffer (300 mM NaCl, 30 mM sodium citrate) for 15 min @RT. Slides were subsequently washed in ddH$_2$O, incubated with 0.1 M Tris buffer (TBS), pH 7.6 for 5 min and 10% BSA in TBS for 10 min at RT. Sections were then incubated with anti-Digoxigenin-Alkaline Phosphatase (anti-sheep F'ab fragments in blocking buffer; 1:100) for 1 h at RT and rinsed with TBS for 5 min. Subsequently, slides were incubated with a chromogenic substrate solution for alkaline phosphatase, BCIP/NBT, rinsed with ddH$_2$O and mounted.

**αSMA–calponin IHC.** Cells were fixed on glass slides with 3.7% paraformaldehyde in PBS for 10 min at room temperature, washed twice with 1xPBS, and permeabilised by incubating in 0.1% triton in PBS for 5 min at room temp, followed by two washes with PBS. Slides were blocked by incubation in 5% FBS in PBS with 0.1% TWEEN 20 for 30 min at RT and then washed twice with PBS. Slides were incubated with primary Ab (anti-calponin (C2687, SIGMA), anti-alpha-SMA (Ab5694, Abcam); 1/1000) for 1 h at RT or overnight at 4 °C. After two PBS washes, slides were incubated in secondary antibody (anti-rabbit alexa-647(4414, Cell Signalling), anti-mouse alexa-488 (A21202, Invitrogen); 1/1000) for 1 h at room temp or overnight at 4 °C, washed twice and then incubated with DAPI to stain nuclei. Slides were imaged using the Leica SP8 laser scanning confocal fluorescence microscope (Leica Microsystems) coupled to a white light laser (WLL), using acquisition software Leica Application Suite X (LAS X).

## Isolation and culture of human aortic vascular smooth muscle cells

Aorta tissue was digested (1 mg/mL collagenase D for 15 min at 37 °C), endothelial cells and adventitia removed and medial tissue further dispersed enzymatically (DMEM/F12, 1% PSF, 165 U/mL Collagenase Type I, 15 U/mL elastase Type III, 0.357 mg/mL soybean trypsin inhibitor for 60 min), vascular smooth muscle cells harvested by centrifugation and seeded (-10,000 cells/cm$^2$) in culture medium (Medium 231 plus Smooth Muscle Growth Supplement (S-007-5), Thermo Fisher Scientific) supplemented with 0.5 mM Ebselen and 1 mM α-tocopherol (SIGMA)[40]. The identity of cells was verified by measuring the expression of VSMC-specific gene and protein markers (Supplementary Fig. 13).

## Cellular studies

Aortic vascular smooth muscle cell (VSMC) or dermal fibroblast H$_2$O$_2$ production was measured using Amplex Red (Thermo Fisher Scientific) with fluorescence measured following the manufacturer's protocol. Lipid peroxidation was measured after loading VSMCs with 1 μM BODIPY 581/591 C$_{11}$ (Thermo Fisher Scientific) with fluorescence monitored by flow cytometry (BD ACCURI C6 plus) on channels FL1-H at 530 nm and FL2-H at 585 nm. Annexin V positivity of VSMCs was measured using a FITC-Annexin V apoptosis detection kit (BD Biosciences) following the manufacturer's protocol. Changes in mitochondrial lipid peroxidation were measured using MitoPerOx (Abcam) detected over time as indicated in figures using the Opera Phenix (Perkin Elmer), with excitation/emission 580 nm/600 nm reflecting the unreacted MitoPerOx compound and excitation/emission 490 nm/520 nm reflecting oxidized MitoPerOx, in a HEPES−Tris buffer: 132 mM NaCl, 10 mM HEPES (pH 7.4), 4.2 mM KCl, 1 mM MgCl$_2$, 1 mM CaCl$_2$, 25 mM D-glucose, 10 mM Tris-HCl (pH 7.4)[41] analysed using Harmony 4.9 Software (Perkin Elmer). Cell viability was measured using alamarBlue™ HS Cell Viability Reagent (Thermo Fisher Scientific) following the manufacturer's protocol. VSMCs were cultured for a week in an antioxidant-free medium or treated with vehicle, 1 mM α-tocopherol (SIGMA), 10 nM MitoQ, 0.5 mM Ebselen, 2.5mM N-acetylcysteine (NAC), 10 mM Na-L-ascorbate, 0.25mM. VSMCs were cultured overnight in antioxidant-free medium with or without a vehicle, 0.25 mM erastin, 25 mM Deferoxamine Mesylate (DFO). H$_2$O$_2$ (0, 50, 100 or 250 μM, 30 min) induced membrane lipid peroxidation in Ficoll purified PBMCs was measured after loading the cells with 1 μM BODIPY 581/591 C$_{11}$ (Thermo Fisher Scientific) with fluorescence monitored by flow cytometry (BD ACCURI C6 plus) on channels FL1-H at 530 nm and FL2-H at 585 nm. An example of the flow cytometry gating strategy used for studies of membrane lipid peroxidation in dermal fibroblasts, VSMCs or PBMCs from patients is shown (Supplementary Fig. 12).

**Radiolabelling of cellular selenoproteins.** Cells were incubated either in standard medium (DMEM plus 10% FBS) supplemented with 10μCi $^{75}$Se (Research reactor facility, University of Missouri) or in methionine-free medium supplemented with 50μCi $^{35}$S-Methionine (Perkin-Elmer, UK), harvested after 48 h, and analysed by SDS−PAGE (10% acrylamide Bis−Tris) and autoradiography[8,42].

**Western blotting of selenoproteins.** Studies were undertaken with antibodies at dilutions specified in Supplementary Table 3. All samples were lysed in RIPA buffer, centrifuged at 12,000 rpm for 10 min at 4 °C and then 2 × SDS−PAGE sample loading buffer was added to the supernatant. Samples were denatured at 95 °C for 5 min and analysed by 10% acrylamide Bis−Tris SDS−PAGE for protein separation using MES-buffer with protein standards appropriate for the molecular weight of selenoproteins of interest[8,42].

**Glutathione peroxidase activity.** Cell lysates (Lysis buffer: 200 mM NaCl, 20 mM Tris−HCl pH7 (NaOH), 1% NP40, 1× protease inhibitor cocktail) were diluted with an equal volume of GPx-assay buffer (50 mM Tris−HCl, 0.5 mM EDTA, pH 8.0) and spun for 10 min at 15,000×g 4 °C to remove insoluble fragments. Diluted samples were distributed in a 96-well plate in triplicate, including a negative control (blank) and a positive control (diluted glutathione peroxidase (5–20 μL of a 0.25 U/ml solution). For each well, 10 μL NADPH assay buffer (5 mM NADPH, 42 mM reduced glutathione, 10 U/mL Glutathione reductase in GPx-assay buffer) was diluted with GPx-assay buffer to a total volume of 200 μL. 2 μL of tBuOOH (fresh 30 mM tert-Butyl hydroperoxide solution made by diluting 4.3 μL Luperox DI to a total volume of 1 mL with water) was added and the decrease in absorption at 340 nm was measured using a kinetic programme taking measurements (at least 10 time points) every 10 s[8,42].

**Thioredoxin reductase activity.** Cell samples in assay buffer (Assay Buffer 5×: 500 mM potassium phosphate, pH 7.0 (61.5 mL K$_2$HPO4 + 38.5 mL KH$_2$PO$_4$), 50 mM EDTA) were sonicated in ice water and centrifuged at 21,000×g for 10 min at 4 °C. Samples (serial dilutions) were distributed in a 96-well plate in triplicate, including a

negative control (blank), a positive control (diluted Thioredoxin Reductase) and with or without 6 μL inhibitor (sodium aurothiomalate (ATM, 20 μM in assay buffer). For each well, further working buffer (1x Assay Buffer 1 mL + NADPH 5 μL (40 mg/mL in $H_2O$) was added to a total volume of 200 μL and the reaction was started by adding 5,5-dithiobis-(2-nitrobenzoic) acid (DTNB) solution (39.6 mg Ellman's Reagent in 1 mL of DMSO). Absorption was measured at 412 nm using a kinetic programme, taking measurements with a 120 s delay (at least 6 time points) every 10 s[8,42].

## Zebrafish models

Zebrafish (*Danio rerio*) embryos obtained from natural spawning were raised and maintained according to EU regulations on laboratory animals (Directive 2010/63/EU). The zebrafish studies were approved by the Body for the Protection of Animals (OPBA) of the University of Milan, Italy (protocol 198283). Depending on the experiments, fish were anesthetized or euthanized with buffered tricaine solution (MS-222, Sigma Aldrich) at 16 or 300 mg/L, respectively. The zebrafish Secisbp2$^{Q333X}$ mutant line (allele designation *sa33758, Tuebingen/AB strain*), generated by the Zebrafish Mutation Project at the Sanger Institute[16] is available from the European (EZRC) or International (ZIRC) resource centre. Tissue samples from Secisbp2$^{Q333X}$ mutant adults were kindly provided by Prof. E. Busch (Sanger Institute, UK). Live embryos were genotyped at 3 days post fertilisation (dpf) by sequencing (Supplementary Fig. 7, Supplementary Table 4). *Secisbp2* morphants (MO) were generated by microinjection of sub-phenotypic doses of translation-blocking (aug-MO, 0.2 pmol/e) and splice-blocking (spl-MO, 0.3 pmol/e) morpholinos (Gene Tools) (Supplementary Fig. 8, Supplementary Table 4). A standard control morpholino (Ct-MO, 0.5 pmol/e) was used to generate control embryos (C) (Supplementary Table 4).

Expression of Secisbp2 and selenoprotein mRNAs in Secisbp2$^{Q333X}$ mutants or *Secisbp2* morphants (Supplementary Figs. 7, 8) were measured by qPCR in pools of 30–40 embryos at 2dpf, with determinations in triplicate. The results were expressed as mean ± standard deviation (SD) and normalized against the elongation factor 1-alpha 1 (*eef1a1*) gene transcript as internal control (Supplementary Table 4).

Measurement of thyroid hormone content or *tshba* transcript levels in adult zebrafish Secisbp2$^{Q333X}$ mutant samples or *Secisbp2* morphant embryos (Supplementary Figs. 7, 8) was undertaken by competitive ELISA (Fish T4/Thyroxine, F10080; Fish T3/Triiodothyronine F10079 kits, LSBio) or qPCR, respectively[43]. To assess the specificity of changes following knockdown with *Secisbp2* MOs, rescue embryos (R) were generated by the microinjection of morpholinos (0.2 pmol/e aug-MO + 0.3 pmol/e spl-MO) and 100 pg/e of *Secisbp2* mRNA at the 1–2-cell stage.

Analyses of reactive oxygen species (ROS) and lipid peroxidation (LPS) were performed using Hydrogen Peroxide Assay (Cayman Chemical) and Lipid Peroxidation (MDA) Assay kits (Sigma Aldrich), as specified by the manufacturer. Measurements were undertaken five times using pools of 100 control, morphant and rescue embryos at 4 dpf (Fig. 4).

Exposure of embryos to $H_2O_2$ (0.5 mM, Sigma), α-tocopherol (1 mM, Sigma), decyl-TPP (Decyltriphenylphosphonium) and MitoQ (1 mM), Erastin (10 mM, STEMCELL Technologies), and desferrioxamine mesylate (DFO, 100 mM, Sigma) was commenced at 3 ($H_2O_2$) or 6 (Erastin) hpf, refreshed twice daily and maintained up to 120 hpf (Figs. 3, 4). Analyses of aortopathy in *Secisbp2* morphants were performed by microinjection of morpholinos into *tg*(*kdrl*:egfp) zygotes, which express a green fluorescent protein in vascular endothelial cells (Fig. 4)[44]. After image acquisition by confocal microscopy, the diameter of the ventral aorta (VA) was measured by ImageJ software and the results were expressed as the mean of VA diameter at three different positions: (1) intersection of VA with pharyngeal vessels, (2) VA midline and (3) junction of VA with bulbus arteriosus (BA). BA diameter

was measured at the widest point of this vascular structure. VA and BA measurements were undertaken in 15 embryos, derived from three independent experiments (Figs. 3, 4).

The survival of wild-type (Secisbp2$^{wt/wt}$; +/+), heterozygous (Secisbp2$^{Q333X/wt}$; +/−) and homozygous (Secisbp2$^{Q333X/Q333X}$; −/−) zebrafish embryos between 1 and 5dpf was calculated in 100–200 embryos derived from three independent matings and expressed as percentage of live embryos (Fig. 3K, O).

Quantitation of apoptotic markers (*p53* and *casp3*) (Figs. 3F and 4F) was undertaken by qPCR, with triplicate determinations using *eef1a1* as an internal control transcript (Supplementary Table 4).

## Mouse model

VSMC-targeted Secisbp2-deficient and control mice were studied following local ethical approval (UK Home Office Project license P452C9545). The Secisbp2$^{flox}$ allele was generated by the International Mouse Phenotyping Consortium (IMPC; details on www.mousephenotype.org[45]. The Myh11-CreERt2, Y chromosome-linked, transgene confers expression of a tamoxifen-inducible Cre recombinase in smooth muscle cells[46,47]. To achieve targeted deletion of *Secisbp2* in VSMCs, Myh11-Cre$^{ERt2}$/Secisbp2$^{flox/flox}$ C57BL/6N male mice (Supplementary Fig. 9B) received intraperitoneal tamoxifen injections (10× 1 mg in corn oil). To assess Secisbp2 expression after recombination, aortas were dissected free from perivascular adipose tissue, endothelial cells removed by gentle scraping, and the adventitial and medial layers separated following brief enzymatic digestion (1 mg/mL Collagenase, Sigma, 1 U/mL Elastase, Worthington). RNA, DNA and protein generated from both adventitial and medial layers were used subsequently for qPCR (Fig. 5A), genotyping (Supplementary Fig. 10) and western blot (Fig. 5B) analyses, using primers (Supplementary Table 5) and antibodies (Supplementary Table 3) specified.

To induce aneurysms, tamoxifen-treated animals were infused with Angiotensin II (0.5 μg/min/kg, Sigma) via an osmotic subcutaneous minipump (ALZET Mini-Osmotic Pump Model 2004) for 28 days. Animals were sacrificed using carbon dioxide after 28 days of Angiotensin II infusion unless they reached a non-humane endpoint or died before. The aorta was dissected, formalin-fixed overnight and examined for macroscopic aneurysm before paraffin embedding. Transverse sections were analysed histologically and immunohistochemically as specified above.

## Statistical analysis

Statistical analyses were performed using GraphPad Prism Version 9.1.2 (GraphPad Software, San Diego, CA, USA, www.graphpad.com) to evaluate normal distribution and variations within and among groups. For all animal studies, the minimal sample size was predetermined by the nature of the experiments. Source data are provided as a Source Data file.

## Reporting summary

Further information on research design is available in the Nature Portfolio Reporting Summary linked to this article.

# Data availability

The genome sequencing data generated in this study are available under restricted access for patient privacy. Sequencing data can only be used to interrogate thyroid or aortopathy-related genes due to the nature of the consent obtained from participants. The processed sequencing data are available upon request from K. Chatterjee (kkc1@medschl.cam.ac.uk), with the period for response to the access request of one calendar month. Access to the UK Biobank genotype and phenotype data is open to all approved health researchers, accessible through https://www.ukbiobank.ac.uk. All other data supporting the findings described in this manuscript are available in the article, Supplementary Information or source data file. Source data are provided with this paper.

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

## Acknowledgements

Our research is supported by a Wellcome Trust Investigator Award (210755/Z/18/Z) and Medical Research Council funding ((MRC_MC_UU_00014/40) to KC, British Heart Foundation (BHF) grants (RG/13/14/30314, RG/20/2/34763, PG/6/24/32090, PG/16/11/32021, PG/13/14/30314; BHF Centre for Research Excellence RE/18/1/34212CH/2000003) and NIHR Senior Investigator award (NF-SI-0616-10036) to MB, Italian Ministry of Health grant (TRANSTIR, # 05C501_2015 and PNRR-MR1-2022-12375726) to LP, Erasmus MC fellowship to WEV and the NIHR Cambridge Biomedical Research Centre. SJJ is supported by an Amsterdam UMC Doctoral Fellowship, and by the Dutch Heart Foundation (Nederlandse Hartstichting 03-007-2022-0035). PTE is supported by the National Institutes of Health (1R01HL092577, 5RO1HL139731, 1R01HL157635), the American Heart Association (18SFRN34110082) and by the European Union (MAESTRIA 965286). JPP is supported by the John S. LaDue Memorial Fellowship for Cardiovascular Research and by the National Institutes of Health (5K08HL159346). Work in the MPM lab is supported by the Medical Research Council UK (MC_UU_00028/4) and by a Wellcome Trust Investigator award (220257/Z/20/Z). Our murine disease model, histopathology and imaging core facilities are supported by MRC Metabolic Diseases Unit [MC_UU_00014/5] and Wellcome Trust funding [208363/Z/17/Z].

## Author contributions

K.C., L.P., M.P.M. and M.B. conceived of and designed the study. E.S., W.E.V., C.M., S.G., C.A., G.L., identified and characterized patients with selenoprotein deficiency and aortopathy, with P.T. analyzing known aortopathy genes in cases. S.J.J., J.P.P. and P.T.E. analysed aortic calibre in SECISBP2 variant carriers in UK Biobank. R.R. and P.C. oversaw cardiological care of patients, with D.G. and J.J.V. undertaking cardiac imaging. S.A.M.N. and R.H. undertook cardiothoracic surgery in patients, with M.G. analyzing aortic histology and N.F. and A.F. performing immunohistochemical studies. E.S., M.A. and S.S. studied patient-derived cells ex vivo. F.M. conducted studies in Secisbp2 mutant (generated by N.W. and E.M.B.-N.) or morpholino knockdown zebrafish models. H.F.J. conducted studies in mice with vascular smooth muscle cell-targeted Secisbp2 deficiency, using a line generated by H.W.-J. and R.R.-S. The manuscript was prepared by E.S., F.M., H.F.J., M.P.M., L.P., M.B. and K.C.

## Competing interests

PTE reports sponsored research support from Bayer AG, IBM Health, Bristol Myers Squibb and Pfizer and has consulted for Bayer AG, Novartis and MyoKardia. LP has consulted for Merck and Sandoz. MPM serves on the SAB of MitoQ Inc. Other authors have no conflicts of interest to declare.

## Additional information

¹Wellcome Trust-MRC Institute of Metabolic Science, University of Cambridge, Cambridge, UK. ²Laboratory of Endocrine and Metabolic Research, Istituto Auxologico Italiano IRCCS, 20149 Milano, Italy. ³Section of Cardiorespiratory Medicine, University of Cambridge, Cambridge, UK. ⁴Department of Internal Medicine and Rotterdam Thyroid Center, Erasmus University Medical Center, Rotterdam, The Netherlands. ⁵Department of Paediatric Endocrinology, Clinica Alemana de Santiago, Vitacura, Chile. ⁶Cardiovascular Disease Initiative, The Broad Institute of MIT and Harvard, Cambridge, MA, USA. ⁷Department of Experimental Cardiology, Amsterdam Cardiovascular Sciences, Amsterdam University Medical Center, Amsterdam, Netherlands. ⁸Wellcome Sanger Institute, Wellcome Genome Campus, Hinxton, UK. ⁹Department of Cardiology, Addenbrookes Hospital, Cambridge, UK. ¹⁰Department of Radiology, Addenbrookes Hospital, Cambridge, UK. ¹¹Cambridge Genomics Laboratory, Addenbrookes Hospital, Cambridge, UK. ¹²Department of Radiology, Erasmus University Medical Center, Rotterdam, The Netherlands. ¹³Department of Pathology, Royal Papworth Hospital, Cambridge, UK. ¹⁴Department of Cardiothoracic Surgery, Royal Papworth Hospital, Cambridge, UK. ¹⁵Department of Cardiothoracic Surgery, Radboud University Medical Center, Nijmegen, The Netherlands. ¹⁶Department of Cardiology, Queen Elizabeth Hospital, Birmingham, UK. ¹⁷Division of Cardiology, University of California San Francisco, San Francisco, CA, USA. ¹⁸Demoulas Center for Cardiac Arrhythmias, Massachusetts General Hospital, Boston, MA, USA. ¹⁹Cardiovascular Research Center, Massachusetts General Hospital, Boston, MA, USA. ²⁰School of Biological and Behavioural Sciences, Queen Mary University of London, London, UK. ²¹MRC Mitochondrial Biology Unit, University of Cambridge, Cambridge, UK. ²²Department of Medical Biotechnologies and Translational Medicine, University of Milan, 20100 Milano, Italy. ²³These authors contributed equally: Erik Schoenmakers, Federica Marelli, Helle F. Jørgensen, W. Edward Visser, Carla Moran. ✉e-mail: kkc1@medschl.cam.ac.uk

