## [Peer Review File · Nature Communications]

REVIEWER COMMENTS

Reviewer #1 (Remarks to the Author):

"Selenoprotein deficiency disorder predisposes to aortic aneurysm formation" carefully follows up on the initial observation of aortic aneurysm in a patient carrying mutations in SECISBP2, a gene involved in the biosynthesis of selenoproteins. Since the original publication of the first patients the authors have reached out to two other patients with this rare genetic disorder, and confirmed the association of aortic aneurysm with mutations in SECISBP2. Then, they created two animal models, a VSMC-specific Secisbp2 knockout in the mouse, and a CRISPR-mutant in zebrafish and showed that both models recapitulate aortopathy. Increased signals from redox-sensitive dyes suggest increased oxidative stress in the aorta - a finding in line with a global reduction of selenoprotein expression (many of which are peroxidases or reductases).

DNA from all four patients was further subjected to exome sequencing. This analysis confirmed the absence of pathogenic variants in 32 known aortopathy genes – supporting the idea that the SECISBP2 variants are pathogenic.

Surgical material from excised aorta and VSMC derived from surgical material showed increased markers of oxidative stress. Treatment of VSMS and patient 2 with lipophilic antioxidants improved the membrane lipid oxidation.

This manuscript carefully explores a novel phenotype of selenoprotein deficiency and even suggests a rationale for future treatment attempts. Thus, this work is important for affected patients, as the paper improved diagnosis, knowledge about pathophysiology, and may help develop a treatment regimen in order to postpone major aortic surgery.

The paper suggests an oxidative cause, but does not elucidate the mechanism of aortopathy.

The conclusions are supported by a huge set of data. And the manuscript is well written and presented.

Questions:

Lipophilic antioxidants were used to reduce oxidative stress in cultured cells or zebrafish embryos. These imply that the authors had GPX4-related mechanisms in mind. Did the authors also use hydrophilic antioxidants? Or why didn't they try hydrophilic antioxidants? Alternatively an inhibitor of ferroptosis might be applied to clarify which branch of oxidants are more crucial for pathology.

The authors observed apoptosis in VSM cells. Based on the rich literature on GPX4 one wonders whether the mechanism of cell death might be ferroptosis. Can the authors provide data in order to distinguish the two forms of cell death and comment in the manuscript? It seems TUNEL+ and AnnV+ are the only basis to call the cell death "apoptotic". It is unclear why the whole term "ferroptosis" is excluded from the manuscript.

In the data mining project using the UK Biobank the authors correlated thoracic aortic diameter in 25 individuals with SECISBP2 LOF variants. It would be nice to provide a table with the actual alleles that were predicted to be LOF together with an allele frequency. For those readers who do not have access to UK Biobank such a table would be very helpful. Maybe the observed pathogenic variants from the study could be included in this table as well in order to get an idea how frequent these alleles are. Such a table might also serve as reference for clinicians seeing a SECISBP2 patient for the first time.

P4 has virtually no selenoprotein expression (except very little TXNRD1). Could the authors comment on how this patient can survive given the severe phenotypes of patients carrying SEPSECS or GPX4 mutations? Is this patient epileptic or shows any sign of neuropathy?

Selenoprotein deficiency in endothelium was reported lethal in mouse models. If space allows this could be discussed in light of the current data.

Data presentation in supplements:

It is unclear how many replicates form the basis for heatmaps depicting selenoprotein expression (several figures including ext data set). If the heatmaps represent more than a single experiment, it would be nice to show all replicates. If these are singular observations, this fact should be stated clearly given the quantitative statement.

Minor:

line 54: should read selenocysteine. SelenoCysteine is not a common abbreviation.

line 122: please mention the number of tested genes (32) as a courtesy to the reader

line 214: the canonical abbreviations are GPX and TXNRD for protein, Gpx and Txnrd for nucleic acids

line 372: von Kossa

Figure 3: proteins should be printed in capital letters. GPX4

Figure 4F: the genotype labels are missing

Ext data set Fig 5: TXRND is unusual, use the canonical abbreviation TNXRD

Western blots are sometimes very tightly cropped. Unclear how good the antibodies really are.

Molecular weight markers should be indicated (e.g. ext data Fig. 2B, Fig. 3, BC, Fig.5B, C).

Reviewer #2 (Remarks to the Author):

Mutation(s) in the SECISBP2 gene results in early onset, progressive dilation of the ascending aorta in patients with a multisystem disorder which is characterized by lack of tissue-specific selenoproteins, cellular oxidative stress caused by the loss of antioxidant selenoenzymes, and abnormal thyroid hormone and selenium levels, which reflect deiodinase selenoenzymes and low levels of selenoproteins. The strength of this well-conducted study lies in its human patient data. Further experiments will provide mechanistic data in cell biology with insight into potential therapeutic options.

1. In VSMC from TAA patients, membrane lipid peroxidation was observed, suggesting that ferroptosis may be involved in medial degeneration. It may be possible for the authors to determine the mitochondrial structural changes in medial cells if increased ferroptosis is observed and aortotomy samples are available. The authors can, at the very least, examine whether mitochondrial ROS levels are elevated in VSMCs from patients and whether the mitochondrial function is affected as a result. A significant increase in ferroptosis may make iron chelators therapeutic for TAA progression and formation.
2. It is stated that VSMC are the cell type that underlies TAA formation due to loss of SECISBP2 activity. In the histochemistry of human and mouse aortic samples, the authors might want to investigate whether the lack of GPX4 activity affects macrophage phenotype or inflammation.
3. In Figure 3F, the aortic samples do not show any TAA. The authors may need to specify when the mice were sacrificed after Ang II infusion.
4. Can the authors test whether a-tocopherol or MitoQ treatment prevents or attenuate the progression of TAA formation in their mouse model?

Minor:

1. Revise the sentence "This phenotype was more pronounced with Secisbp2 morpholino knockdown in embryos, perhaps due to genetic compensation in the mutant model" for clarity.
2. Label Figure 3F.

Reviewer #3 (Remarks to the Author):

The manuscript by Schoenmakers, Marelli, Jorgensen, Visser, and Moran et al, demonstrates that selenoprotein deficiency disorder predisposes humans to aortic aneurysm. They then go on to model this in zebrafish and mouse. The strength of the manuscript is the finding that selenoprotein deficiency disorder causes aneurysm, which, according to my brief google search, has not been demonstrated before in the literature. The overall weaknesses of the study are (1) incomplete animal modeling and phenotyping, and (2) a lack of mechanistic insight.

Major Weakness/Concern:

Adult zebrafish phenotype: The diameter of the VA is not normalized to fish body length or body weight (either would work and should be reported). If control body length and weight are not different between control and mutant animals, then normalization is not an issue. The diameter of the BA, equivalent of the aortic root in mammals, should also be determined and reported. The sample size in B should also be increased for the VA and BA measurements as the increase is relatively small.

Use of Morpholinos in Zebrafish and Embryonic Phenotype Interpretation: The discrepancy in aortic dilation phenotypes between *secisbp2* morphant and mutant embryos is concerning (I'm assuming there is no dilation during embryonic stages in the mutant animals). While the authors try to mitigate this concern by rescuing the morpholino with mRNA injection, this approach is not accepted in the field. Overall, the use of morpholinos is the least convincing part of the zebrafish data and I suggest removing these data and focusing more on the genetic mutant.

It is possible that *secisbp2* is maternally deposited, which could theoretically protect *secisbp2* mutants from aortic dilation. However, this explanation is unlikely as phenotypic protection from most maternal RNAs/proteins is short-lived with phenotypes revealing themselves on day 3 and beyond. The authors should examine heterozygous and homozygous *secisbp2* mutants for VA and BA dilation at 5 days. If no dilation is seen, then I would interpret this result to mean that selenoprotein deficiency results in chronic abnormalities that result in aneurysmal dilation over time. They should also treat a clutch of embryos from heterozygous incrosses with H₂O₂ to learn if the hets and/or null embryos are sensitized to VA/BA dilation compared to WT controls. The embryos should be genotyped after imaging.

How does the mouse phenotype compare to the zebrafish? The authors show that VSMC-specific deletion of *Secisbp2* in adult mice results in early death caused by TAA rupture after angiotensin II exposure. Does that mean that the hets and homozygous null animals do not show TAA without angiotensin II treatment? A specific statement should be made regarding the phenotype in the absence of angII. Have the authors activated Cre earlier (P1-3 stages?) to see if TAA occurs in adult mice without angII?

Rescue experiments that suggest causative mechanisms: Can the authors rescue TAA or stop TAA progression in zebrafish or mice by treatment with mitoQ or α -Tocopherol? Other strategies that might rescue or slow progression? More mechanistic insights should be gained from the animal models.

Minor Concerns and Suggestions:

Consider adding a few sentences at the end of the Introduction to say what is known about links between selenoprotein deficiency and cardiovascular disease – especially TAA. If nothing is known or if it has never been investigated, then that would be important too.

Line 75: It is not clear what “different genes” means. Do the authors mean the “different genes” they list in the sentences above or do they mean genes other than the ones they listed above? This should be clarified.

Showing that the VSMCs of Patient 1 and 2 show deficiencies in selenoproteins seems like an important component to the story. As such, the authors might want to incorporate those data into main figure. 1 instead of in the extended figures.

Reviewer #4 (Remarks to the Author):

In this paper, the authors describe TAA in patients with SECISBP2 variants. The findings are supported by studies in zebrafish and mice. Overall, I find the results interesting, but the paper very short, which limits the discussion and other parts of the paper, and makes it difficult to read. E.g. there is no methods section, which makes the paper less reader-friendly. So the paper needs to be re-organized and re-written.

Specific points:

The phenotype in the patients here compared to previously reported patients is only discussed briefly and should be extended. I lack a paragraph about the importance of followup of patients re aorta.

The results section is quite disorganized. I suggest to write the genetics results in a separate section. Also a separate section for results of selenoprotein in VMSCs.

The first sentence of the abstract is very general, should be more specific.

In the description of diseases associated with SECISBP2 variants the OMIM number is missing. I miss a description of the variants found in the patients in the text. It is not sufficient to describe them in the figures.

The nomenclature for gene variants should be updated, e.g. variant should be used instead of mutation. Also, the correct nomenclature for sequence variants and their consequences should be used.

In supplementary methods, a description of data filtering is missing. How were the SECISBP2 variants identified?

Also, it is mentioned that the lack of TAA in previously reported patients could be due to age difference and/or specific variants. This should be elaborated.

Supplementary p 7: VSMC abbreviation should be explained

REVIEWER COMMENTS

Reviewer #1 (Remarks to the Author):

Questions:

Lipophilic antioxidants were used to reduce oxidative stress in cultured cells or zebrafish embryos. These imply that the authors had GPX4-related mechanisms in mind. Did the authors also use hydrophilic antioxidants? Or why didn't they try hydrophilic antioxidants? Alternatively an inhibitor of ferroptosis might be applied to clarify which branch of oxidants are more crucial for pathology.

We thank the Reviewer for this suggestion and have now included additional data, showing that other antioxidants (ebselen, N-acetylcysteine and sodium ascorbate) are much less effective in preventing oxidative damage or preventing apoptosis of aortic vascular smooth muscle cells from patients (Extended Data Fig 7).

We have evaluated the role of ferroptosis, testing erastin (an activator of this iron-dependent cell death pathway and desferrioxamine an iron chelator that decreases ferroptosis in both aortic VSMCs from patients (Fig 1G and 1H) and *Secisbp2* mutant (Fig 2L,M,N,O) or knockdown (Extended Data Fig 10J-L) zebrafish embryos as described in more detail below.

The authors observed apoptosis in VSM cells. Based on the rich literature on GPX4 one wonders whether the mechanism of cell death might be ferroptosis. Can the authors provide data in order to distinguish the two forms of cell death and comment in the manuscript? It seems TUNEL+ and AnnV+ are the only basis to call the cell death "apoptotic". It is unclear why the whole term "ferroptosis" is excluded from the manuscript.

We thank the Reviewer for this very helpful suggestion and have now addressed this question with additional experiments and data. We tested erastin, a small molecule which blocks cystine uptake (System xc⁻), thereby activating the ferroptotic cell death pathway. In keeping with their lack of GPX4 (Extended Data Fig 2), the aortic VSMCs from patients are more susceptible to erastin compared to controls, with increased oxidative damage and reduced cell viability. Desferrioxamine (DFO), an iron chelator, prevented these deleterious effects (Fig 1G, Fig 1H). Erastin also evoked ventral aortic dilatation and reduced survival of *Secisbp2* mutant zebrafish embryos, with desferrioxamine preventing these deleterious effects (Fig 2L, 2M, 2N, 2O); similar observations were made in *Secisbp2* morpholino knockdown (Extended Data Fig 10J-L) zebrafish embryos.

In the data mining project using the UK Biobank the authors correlated thoracic aortic diameter in 25 individuals with SECISBP2 LOF variants. It would be nice to provide a table with the actual alleles that were predicted to be LOF together with an allele frequency. For those readers who do not have access to UK Biobank such a table would be very helpful. Maybe the observed pathogenic variants from the study could be included in this table as well in order

to get an idea how frequent these alleles are. Such a table might also serve as reference for clinicians seeing a SECISBP2 patient for the first time.

We thank the Reviewer for this suggestion and now include Excel Files in Supplementary Information specifying the nature of LoF *SECISBP2* variants in UKBiobank and their frequency in this database. We have also tabulated the frequency of the known, pathogenic, SECISBP2 variants from P1,P2,P3,P4 in UK Biobank in a tab marked SECISBP2_patient_var_counts_UKB.

P4 has virtually no selenoprotein expression (except very little TXNRD1). Could the authors comment on how this patient can survive given the severe phenotypes of patients carrying SEPSECS or GPX4 mutations? Is this patient epileptic or shows any sign of neuropathy?

We acknowledge that, when tested by radiolabelling or western blotting, dermal fibroblasts from P4 express very few selenoproteins. However, primary T lymphocytes from this patient (Extended Data Fig 5A) do express selenoproteins other than TXNRD1, suggesting that selenoprotein deficiency varies between different tissues or cell types, possibly accounting for her survival.

As documented in her Case History (Extended Data Page 11), this patient has signs of early muscular dystrophy but no neurological abnormalities (e.g. epilepsy or neuropathy). However, such paucity or even absence of neurological phenotypes is also a recognised feature of other SECISBP2 deficiency cases and is indeed in contrast to the human SEPSECS deficiency disorder which is associated with cortical and cerebellar atrophy (Schoenmakers, Int J Mol Sci 2021 PMID: 34884733).

Selenoprotein deficiency in endothelium was reported lethal in mouse models. If space allows this could be discussed in light of the current data.

Our mouse model was genetically engineered to successfully abrogate SECISBP2 expression selectively in the medial aortic layer (Fig 3A) and we surmise that, similar to preserved, normal SECISBP2 mRNA expression in the adventitial layer (Fig 3A), endothelial SECISBP2 and therefore selenoprotein expression is preserved.

Data presentation in supplements:

It is unclear how many replicates form the basis for heatmaps depicting selenoprotein expression (several figures including ext data set). If the heatmaps represent more than a single experiment, it would be nice to show all replicates. If these are singular observations, this fact should be stated clearly given the quantitative statement.

The heatmaps presented were not generated from a single measurement. In Supplementary Information we have now included a Primary Data file which contains the data from which heat maps (Extended Data Fig 2, Fig 3, Fig 8, Fig 9) were generated.

Minor:

line 54: should read selenocysteine. SelenoCysteine is not a common abbreviation.

We have corrected SelenoCysteine to selenocysteine.

line 122: please mention the number of tested genes (32) as a courtesy to the reader

We have specified 32 known aortopathy genes were tested (line 120).

line 214: the canonical abbreviations are GPX and TXNRD for protein, Gpx and Txnrd for nucleic acids

We have changed nomenclature to GPX and TXNRD (line 231), to denote proteins rather than nucleic acids.

line 372: von Kossa

We have amended Von Kossa to von Kossa in the legend to Figure 1.

Figure 3: proteins should be printed in capital letters. GPX4

We have amended the nomenclature of selenoproteins in Figure 3B and Extended Data Fig 2A,B, Fig 5A.

Figure 4F: the genotype labels are missing

We thank the Reviewer and surmise that they are referring to Fig 3F. The genotype labels have been added to Figure 3F.

Ext data set Fig 5: TXRND is unusual, use the canonical abbreviation TNXRD

Throughout the manuscript we have adopted TXNRD as the nomenclature for thioredoxin reductase, corresponding to its entry in GeneCards.

Western blots are sometimes very tightly cropped. Unclear how good the antibodies really are. Molecular weight markers should be indicated (e.g. ext data Fig. 2B, Fig. 3, BC, Fig.5B, C).

We recognise that the western blots are cropped. Accordingly, in Supplementary Information, we now include uncropped western blots, annotated with molecular weight markers denoting bands corresponding to specific selenoproteins, which are boxed.

Reviewer #2 (Remarks to the Author):

Mutation(s) in the SECISBP2 gene results in early onset, progressive dilation of the ascending aorta in patients with a multisystem disorder which is characterized by lack of tissue-specific selenoproteins, cellular oxidative stress caused by the loss of antioxidant selenoenzymes, and abnormal thyroid hormone and selenium levels, which reflect deiodinase selenoenzymes and low levels of selenoproteins. The strength of this well-conducted study lies in its human patient data. Further experiments will provide mechanistic data in cell biology with insight into potential therapeutic options.

1. In VSMC from TAA patients, membrane lipid peroxidation was observed, suggesting that ferroptosis may be involved in medial degeneration. It may be possible for the authors to determine the mitochondrial structural changes in medial cells if increased ferroptosis is observed and aortotomy samples are available. The authors can, at the very least, examine whether mitochondrial ROS levels are elevated in VSMCs from patients and whether the mitochondrial function is affected as a result. A significant increase in ferroptosis may make iron chelators therapeutic for TAA progression and formation.

We are grateful to the Reviewer for these suggestions and have undertaken additional experiments and included data to address these issues.

Using MitoPeroX, a mitochondria-targeted fluorescent probe which detects lipid peroxidation in mitochondrial membranes, we find that such oxidative damage is markedly increased in aortic vascular smooth muscle cells from patients (P1, P2) compared to controls (C1, C2, C3) (Extended Data Fig 6). This finding indicates that oxidative damage to mitochondria in patient's cells is also a feature of this disorder.

This observation has now been added to the text in Results (Page 6, lines 136-137) and Discussion (Page 9, line 219).

We tested erastin, a small molecule which blocks cystine uptake (System xc⁻), thereby activating the ferroptotic cell death pathway. In keeping with their lack of GPX4 (Extended Data Fig 2), the aortic VSMCs from patients are more susceptible to erastin compared to controls, with increased oxidative damage and reduced cell viability. Desferrioxamine (DFO), an iron chelator, prevented these deleterious effects (Fig 1G, Fig 1H). Erastin also evoked ventral aortic dilatation and reduced survival of *Secisbp2* mutant zebrafish embryos, with desferrioxamine preventing these deleterious effects (Fig 2L, 2M, 2N, 2O); similar observations were made in *Secisbp2* morpholino knockdown (Extended Data Fig 10J-L) zebrafish embryos.

These findings have now been included in the text in Results (Page 6, lines 139-142; page 7, lines 163-173) and Discussion (Page 9, lines 204-209; line 222. Page 10, line 240-241).

2. It is stated that VSMC are the cell type that underlies TAA formation due to loss of SECISBP2 activity. In the histochemistry of human and mouse aortic samples, the authors might want to investigate whether the lack of GPX4 activity affects macrophage phenotype or inflammation.

We acknowledge that inflammation can contribute to TAA formation. To address this we undertook additional experiments and have included this data. In Extended Data Figure 13, immunohistochemistry, using anti-CD68 antibody, of a region of aortic degeneration from patient P1, shows no macrophage inflammatory infiltrate. Similarly, immunohistochemistry of mouse thoracic aorta, using anti-MAC-3 antibody, shows macrophage infiltration in both wild type and *Secisbp2* knockout mice that is comparable to that seen in wild type, angiotensin II-infused, control animals.

These observations have now been added to the text in Results (page 6, line 132; page 8, lines 185-187)

3. In Figure 3F, the aortic samples do not show any TAA. The authors may need to specify when the mice were sacrificed after Ang II infusion.

As specified in the protocol shown in Extended Data Figure 11B, mice were sacrificed and analysed 28 days after infusion of Angiotensin II.

4. Can the authors test whether a-tocopherol or MitoQ treatment prevents or attenuate the progression of TAA formation in their mouse model?

Due to the COVID pandemic, breeding of mouse lines used in this study was curtailed. Accordingly, it has not been possible to undertake the experiments suggested above in the *mouse* model, to fit within the three-month timeframe to revise this manuscript, suggested by the Editor. However, we have undertaken exactly these studies in both *Secisbp2* mutant and morpholino knockdown models. Both α -tocopherol and MitoQ prevent aortic dilatation of either *Secisbp2*^{Q333X/Q333X} mutant (Fig 2G-O) or *Secisbp2* morpholino knockdown zebrafish embryos (Extended Data Fig 10G-I).

These findings have now been included in the text in Results (Page 7, lines 163-169) and Discussion (Page 9, lines 204-209; Page 10, lines 223-226. Page 10, line 240-241).

Minor:

1. Revise the sentence “This phenotype was more pronounced with *Secisbp2* morpholino knockdown in embryos, perhaps due to genetic compensation in the mutant model” for clarity.

In data from additional experiments included in the revised manuscript we observe very similar deleterious effects of oxidative stress (H₂O₂) or activation of the ferroptotic cell death pathway in either *Secisbp2*^{Q333X/Q333X} mutant (Fig 2G-O) or *Secisbp2* morpholino knockdown zebrafish embryos (Extended Data Fig 10G-L).

Accordingly, the revised manuscript no longer includes this sentence.

2. Label Figure 3F.

We thank the Reviewer and have added genotype labels to Figure 3F.

Reviewer #3 (Remarks to the Author):

The manuscript by Schoenmakers, Marelli, Jorgensen, Visser, and Moran et al, demonstrates that selenoprotein deficiency disorder predisposes humans to aortic aneurysm. They then go on to model this in zebrafish and mouse. The strength of the manuscript is the finding that selenoprotein deficiency disorder causes aneurysm, which, according to my brief google search, has not been demonstrated before in the literature. The overall weaknesses of the study are (1) incomplete animal modeling and phenotyping, and (2) a lack of mechanistic insight.

Major Weakness/Concern:

Adult zebrafish phenotype: The diameter of the VA is not normalized to fish body length or body weight (either would work and should be reported). If control body length and weight are not different between control and mutant animals, then normalization is not an issue. The diameter of the BA, equivalent of the aortic root in mammals, should also be determined and reported. The sample size in B should also be increased for the VA and BA measurements as the increase is relatively small.

In the legend to Fig2, it is now stated that measurements of ventral aorta (VA) (Fig 2B) and bulbus arteriosus (BA) diameters (Fig 2C) in six month old, adult wild-type (*Secisbp2*^{wt/wt}; +/+), heterozygous (*Secisbp2*^{Q333X/wt}; +/-) and homozygous (*Secisbp2*^{Q333X/Q333X}; -/-) mutant, are adjusted for body size.

In Supplementary Information, the tab in the raw data Excel file corresponding to Fig2B & 2C indicates there was no significant difference in size of wild type, heterozygous or homozygous adult, mutant animals.

We acknowledge that the increase in VA diameter in *Secisbp2*^{Q333X/Q333X}; -/- animals is small but it remains significant. Due to the limited timeframe (three months) permitted by the Editor for revision of this manuscript, it has not been possible to generate larger numbers of six month old adult, mutant zebrafish for analysis.

Use of Morpholinos in Zebrafish and Embryonic Phenotype Interpretation: The discrepancy in aortic dilation phenotypes between *secisbp2* morphant and mutant embryos is concerning (I'm assuming there is no dilation during embryonic stages in the mutant animals). While the authors try to mitigate this concern by rescuing the morpholino with mRNA injection, this approach is not accepted in the field. Overall, the use of morpholinos is the least convincing part of the zebrafish data and I suggest removing these data and focusing more on the genetic mutant.

We fully recognise potential limitations of the *Secisbp2* morpholino knockdown zebrafish embryo model. Exactly as specified by this Reviewer (see below), we have now undertaken experiments in *Secisbp2* mutant embryos and observe that either oxidative stress (H₂O₂) or exposure to Erastin (an activator of ferroptosis) are associated with significant aortic dilatation and reduced survival of homozygous *Secisbp2*^{Q333X/Q333X}; -/- mutant zebrafish embryos at 5dpf, with antioxidants (α -tocopherol, MitoQ) or desferrioxamine (an iron chelator), preventing these deleterious effects (Fig 2G-O).

Exposure to H₂O₂ or Erastin also caused ventral aortic dilatation in *Secisbp2* morpholino knockdown zebrafish embryos, with antioxidants (α -tocopherol, MitoQ) or desferrioxamine (an iron chelator), preventing these deleterious effects. Accordingly, rather than discarding this model, we have included this data in Supplementary Information (Extended Data, Fig 10G-O).

It is possible that *secisbp2* is maternally deposited, which could theoretically protect *secisbp2* mutants from aortic dilation. However, this explanation is unlikely as phenotypic protection from most maternal RNAs/proteins is short-lived with phenotypes revealing themselves on

day 3 and beyond. The authors should examine heterozygous and homozygous *secisbp2* mutants for VA and BA dilation at 5 days. If no dilation is seen, then I would interpret this result to mean that selenoprotein deficiency results in chronic abnormalities that result in aneurysmal dilation over time. They should also treat a clutch of embryos from heterozygous incrosses with H₂O₂ to learn if the hets and/or null embryos are sensitized to VA/BA dilation compared to WT controls. The embryos should be genotyped after imaging.

We are very grateful to the Reviewer for suggesting such experiments and, as described above, have undertaken them and included this data in the revised manuscript (Fig 2G-O).

How does the mouse phenotype compare to the zebrafish? The authors show that VSMC-specific deletion of *Secisbp2* in adult mice results in early death caused by TAA rupture after angiotensin II exposure. Does that mean that the hets and homozygous null animals do not show TAA without angiotensin II treatment? A specific statement should be made regarding the phenotype in the absence of angII. Have the authors activated Cre earlier (P1-3 stages?) to see if TAA occurs in adult mice without angII?

Since many groups have reported that angiotensin II infusion in mice can model thoracic aortic aneurysm formation (Reviewed in PMID 28539494), and can recapitulate human thoracic aneurysm disorders (Transforming growth factor β -SMAD3 signalling, PMID 29519942; matrix metalloproteinase 2 PMID 25657308), we chose to use this experimental paradigm to determine whether VSMC-specific selenoprotein deficiency can mediate this phenotype. We have not conditionally deleted *secisbp2* at earlier stages (e.g. P1-3) in mice, but would be concerned that this could be associated with another life-limiting abnormality outside the vascular system, similar to known embryonic lethality of germline, murine *secisbp2* knockout (PMID 24274065).

Rescue experiments that suggest causative mechanisms: Can the authors rescue TAA or stop TAA progression in zebrafish or mice by treatment with mitoQ or α -Tocopherol? Other strategies that might rescue or slow progression? More mechanistic insights should be gained from the animal models.

We thank the Reviewer for this valuable suggestion and have now undertaken experiments and incorporated additional data, to address this directly. Both α -tocopherol and MitoQ prevent aortic dilatation of either *Secisbp2*^{Q333X/Q333X} mutant (Fig 2G-O) or *Secisbp2* morpholino knockdown zebrafish embryos (Extended Data Fig 10G-I). These findings have now been included in the text in Results (Page 7, lines 163-169) and Discussion (Page 9, lines 202-209; lines 219-222. Page 10, line 240-242).

We also tested erastin, a small molecule which blocks cystine uptake (SystemX_c⁻), thereby activating the ferroptotic cell death pathway. In keeping with their lack of GPX4 (Extended Data Fig 2), the aortic VSMCs from patients are more susceptible to erastin compared to controls, with increased oxidative damage and reduced cell viability. Desferrioxamine, an iron chelator, prevented these deleterious effects (Fig 1G, Fig 1H). Erastin also evoked ventral aortic dilatation and reduced survival of *Secisbp2*^{Q333X/Q333X} mutant zebrafish embryos, with

desferrioxamine preventing these deleterious effects (Fig 2L, 2M, 2N, 2O); similar observations were made in *Secisbp2* morpholino knockdown (Extended Data Fig 10J-L) zebrafish embryos.

These findings have now been included in the text in Results (Page 7, lines 163-169) and Discussion (Page 9, lines 204-209; Page 10, lines 223-226. Page 10, line 240-241).

Minor Concerns and Suggestions:

Consider adding a few sentences at the end of the Introduction to say what is known about links between selenoprotein deficiency and cardiovascular disease – especially TAA. If nothing is known or if it has never been investigated, then that would be important too.

We thank the Reviewer for this suggestion and have added the following sentence to the end of the Introduction

Although individual (e.g. Selenoprotein P) or a subset (e.g. endoplasmic reticulum resident) of selenoproteins have been linked to adverse cardiovascular outcomes (Schomburg Int J Mol Sci 2021) or cardioprotection (Rocca, Cellular & Molecular Life Sciences 2019) respectively, they have not been associated with vascular pathology. (Page 4, lines 86-88).

Line 75: It is not clear what “different genes” means. Do the authors mean the “different genes” they list in the sentences above or do they mean genes other than the ones they listed above? This should be clarified.

We have modified this sentence to read “defects in these and other genes” (Page 3, line 72).

Showing that the VSMCs of Patient 1 and 2 show deficiencies in selenoproteins seems like an important component to the story. As such, the authors might want to incorporate those data into main figure. 1 instead of in the extended figures.

As requested by this Reviewer, we have undertaken additional experiments which have provided mechanistic insights, including susceptibility of aortic VSMCs from patients to ferroptosis (an iron-dependent cell death pathway). We have included these important new observations into main Fig 1 (panels G and H). Accordingly, we have respectfully retained selenoprotein expression data from P1 and P2 in Extended Data Fig 2.

Reviewer #4 (Remarks to the Author):

In this paper, the authors describe TAA in patients with SECISBP2 variants. The findings are supported by studies in zebrafish and mice. Overall, I find the results interesting, but the paper very short, which limits the discussion and other parts of the paper, and makes it difficult to read. E.g. there is no methods section, which makes the paper less reader-friendly. So the paper needs to be re-organized and re-written.

We recognise that this paper contains a lot of information, comprising a new phenotype (aortic aneurysm) in a complex human disorder (selenoprotein deficiency), together with supporting information mechanistic insights from two complementary animal models (global and tissue-selective selenoprotein deficiency in zebrafish and mice). Accordingly, the Methods section is necessarily extensive and long (ten pages) and we respectfully suggest that it is better included in Supplementary Information rather than the main manuscript.

However, we have reorganised and rewritten other parts of the paper to take valuable suggestions from this Reviewer into account, as detailed below.

Specific points:

The phenotype in the patients here compared to previously reported patients is only discussed briefly and should be extended. I lack a paragraph about the importance of follow-up of patients re aorta.

In the revised main manuscript, we now report *SECISBP2* variant genotypes and aortic phenotypes. In Extended Data page 11 we have now included detailed Case Histories of patients (P1 to P4), describing previously reported phenotypes, their clinical progress and details of ongoing aortic surveillance.

The results section is quite disorganized. I suggest to write the genetics results in a separate section.

As identification of variants in the *SECISBP2* gene for some patients has been reported previously (P2, PMID: 21084748; P3, PMID: 19602558), we respectfully suggest that describing this information in the Case Description part of Results is more appropriate.

Also a separate section for results of selenoprotein in VMSCs.

Similarly, in one patient selenoprotein expression in his cells has been reported previously (P2, PMID: 21084748). In other patients (P1, P3, P4) we respectfully suggest that, since the selenoprotein expression profiles are included with *SECISBP2* variant genotypes in Extended Data (Fig 2, Fig 3, Fig 5), it remains appropriate to describe them together.

The first sentence of the abstract is very general, should be more specific.

We have rephrased the first sentence of the abstract as follows:

Aortic aneurysms, which may dissect or rupture acutely and be lethal, can be a part of multisystem disorders that have a heritable basis. (Page 3, lines 50-51).

In the description of diseases associated with *SECISBP2* variants the OMIM number is missing. I miss a description of the variants found in the patients in the text. It is not sufficient to describe them in the figures.

We thank the Reviewer for this suggestion and have now included the OMIM number (607693) when first describing this disorder in the Introduction. (Page 4, line 80).

The nomenclature for gene variants should be updated, e.g. variant should be used instead of mutation. Also, the correct nomenclature for sequence variants and their consequences should be used.

We have substituted the term variant for mutation throughout the manuscript. In addition we have now used the correct nomenclature when describing variants (e.g. c.382 C>T) and their consequences (e.g. p.R182X) and included these in descriptions of patients in the main manuscript (Page 4, lines 95-96; Page 5, line 100; line 106; lines 114-115).

In supplementary methods, a description of data filtering is missing. How were the SECISBP2 variants identified?

Extended Data (page 5) contains the following paragraph which describes how LOF variants in SECISBP2 were identified.

Variants were annotated using the Loss-of-Function Transcript Effect Estimator (LOFTEE⁸) plug-in implemented in the Variant Effect Predictor (VEP; v.105)⁹ (<https://github.com/konradjk/loftee>). LOFTEE was implemented to identify high-confidence LOF variants for the canonical transcript of *SECISBP2*, which include frameshift indels, stop-gain variants and splice site disrupting variants. We further annotated variants with continental allele frequencies from gnomAD⁸, identifying a minor allele frequency (MAF) thresholds (MAF <0.1% in exome and gnomAD datasets) for inclusion of LOF variants. After annotation, we identified 25 carriers of rare (MAF<0.1%) *SECISBP2* LOF variants amongst the participants.

Also, it is mentioned that the lack of TAA in previously reported patients could be due to age difference and/or specific variants. This should be elaborated.

In the Discussion, we have rephrased this paragraph (page 8-9, lines 196-201) to read as follows:

Aortic dilatation has not yet been documented in nine other known cases of selenoprotein deficiency, but as other patients harbor different, pathogenic *SECISBP2* variants or were described in childhood or early adolescence (age 2-14yrs)¹⁸, it is conceivable that the development of aortopathy may vary, depending either on underlying *SECISBP2* variant genotype or manifest at an older age due to oxidative tissue damage being cumulative.

Supplementary p 7: VSMC abbreviation should be explained

We have now specified that this abbreviation corresponds to vascular smooth muscle cell.

REVIEWERS' COMMENTS

Reviewer #1 (Remarks to the Author):

New data has been generated and additional data has been included in the supplementary data section. Small amendmends and correccions have been incorporated. The revised manuscript satisfies my questions and comments. Congratulations, this is a significant piece of work spanning several preclinical models, patients, and population data bases.

Reviewer #2 (Remarks to the Author):

The authors have answered the reviewers' questions. Publication of the manuscript is recommended.

Reviewer #3 (Remarks to the Author):

In the revised manuscript, "Selenoprotein deficiency disorder predisposes to aortic aneurysm formation", the authors have adequately and reasonably addressed the comments from the original review. The new data added, specifically in respect to the animal models, gives further mechanistic insight into how selenoprotein deficiency might cause aortic dilation.

Reviewer #4 (Remarks to the Author):

I find that the manuscript has been corrected appropriately according to the reviewers' comments